# Effectiveness of *Pseudomonas aeruginosa* type VI secretion system relies on toxin potency and type IV pili-dependent interaction

**Marta Rudzite[1], Sujatha Subramoni[2], Robert G. Endres[3], Alain Filloux [1,2]***

**1** MRC Centre for Molecular Bacteriology and Infection, Department of Life Sciences, Imperial College London, London, United Kingdom, **2** Singapore Centre for Environmental Life Sciences Engineering, Nanyang Technological University, Singapore, **3** Centre for Integrative Systems Biology and Bioinformatics, Department of Life Sciences, Imperial College London, London, United Kingdom

* a.filloux@imperial.ac.uk, a.filloux@ntu.edu.sg

**Data Availability Statement:** All relevant data are within the manuscript and its Supporting information files.

## Abstract

The type VI secretion system (T6SS) is an antibacterial weapon that is used by numerous Gram-negative bacteria to gain competitive advantage by injecting toxins into adjacent prey cells. Predicting the outcome of a T6SS-dependent competition is not only reliant on presence-absence of the system but instead involves a multiplicity of factors. *Pseudomonas aeruginosa* possesses 3 distinct T6SSs and a set of more than 20 toxic effectors with diverse functions including disruption of cell wall integrity, degradation of nucleic acids or metabolic impairment. We generated a comprehensive collection of mutants with various degrees of T6SS activity and/or sensitivity to each individual T6SS toxin. By imaging whole mixed bacterial macrocolonies, we then investigated how these *P. aeruginosa* strains gain a competitive edge in multiple attacker/prey combinations. We observed that the potency of single T6SS toxin varies significantly from one another as measured by monitoring the community structure, with some toxins acting better in synergy or requiring a higher payload. Remarkably the degree of intermixing between preys and attackers is also key to the competition outcome and is driven by the frequency of contact as well as the ability of the prey to move away from the attacker using type IV pili-dependent twitching motility. Finally, we implemented a computational model to better understand how changes in T6SS firing behaviours or cell-cell contacts lead to population level competitive advantages, thus providing conceptual insight applicable to all types of contact-based competition.

## Author summary

The Type VI Secretion System (T6SS) was discovered in Gram-negative bacteria and is a contact-dependent molecular nanomachine which injects antimicrobial toxins into competitors. The T6SS activity provides a competitive edge for an organism to prevail in a given niche. The number and biochemical activity of toxins injected could vary from one bacterial species to the other. The opportunistic pathogen *Pseudomonas aeruginosa* is quite prolific with more than 20 characterized and distinct T6SS toxins. Here we

**Funding:** This work was supported by a MRC grant n° MR/S02316X/1 to AF. MR is supported by a BBSRC DTP studentship (reference 2133361). The funders had no role in study design, data collection and analysis, decision to publish, or preparation of the manuscript.

**Competing interests:** The authors have declared that no competing interests exist.

experimentally addressed the importance in the degree of T6SS activity and made the demonstration that every single of the many *P. aeruginosa* T6SS toxin counts. There are no redundancies and instead on occasion synergies, which support the effectiveness of injecting a cocktail of toxins rather than a subset of specific ones. Since contact is a key factor for T6SS delivery, we further observed that preys able to use surface motility skills to run away from bacterial attackers have better chances to multiply and survive. Furthering experimental approaches, we used computational modelling to place these data in the context of a mixed *P. aeruginosa* population. This way we contribute understanding to how highly local contact-based interactions between individuals shape the structure of whole bacterial communities.

## Introduction

Bacteria thrive by adapting to a wide variety of ecological niches [1]. This includes commensals that are a part of the host microbiota [2] or bacterial pathogens colonising a host [3]. In any environment, resources can be scarce and the competition for survival a serious challenge. The structure of a polymicrobial population steadily establishing in a niche relies on competition [4] and cooperation [5]. For example, cooperation arises from the ability of a species to catabolise complex nutrients sources that is then used by other species. In contrast, competition aims to eliminate cheaters and foes and relies on a variety of fighting strategies [6]. Polymicrobial communities can be highly complex, and for example up to 40,000 species can coexist within the human gut [2]. These populations can adopt a biofilm lifestyle which contributes stability and resilience [7]. Due to the complexity of cell-cell interactions, the development of polymicrobial communities is notoriously difficult to predict but has many implications in ecology, industry, and medicine [8].

The outcome of a competition is dependent on various skills that bacteria have acquired during evolution [1]. Some species can be very effective at capturing rare elements such as iron, by producing high affinity siderophores [9] or proteins able to recapture iron from transferrin or lactoferrin [10]. The depletion of iron is detrimental to species less able to capture it. This strategy to starve others to their death is more reminiscent of a siege in terms of combat, whereas direct competition strategies involve frontal assault. For long it has been known that bacteria such as *Escherichia coli* release antibacterial toxins named colicins that are able to penetrate and kill related species [11]. A contact-dependent mechanism was later discovered which allows specific delivery of toxins, such as tRNAse, into related preys [12]. This system is called CDI for contact-dependent inhibition [13] and uses specific receptors at the prey-cell surface to make contact [14]. Interestingly, a contact-dependent system with much broader impact was discovered in *Pseudomonas aeruginosa* [15]. The system delivers a cocktail of antibacterial toxins with various biochemical activities and does not seem to display specificity towards a particular kind of bacterial prey [16,17]. It can inject toxic effectors into prokaryotes, but can also target fungi [18] and other eukaryotic cells such as amoeba or host macrophages [19,20]. It is called the type VI secretion system (T6SS) and is broadly conserved in Gram-negative bacteria [21–23]. The T6SS involves a contractile sheath, TssBC, which is full of Hcp rings loaded with antibacterial toxins. In recent years it has been found that other secretion systems, including the type IV secretion system (T4SS) [24] or the type VII secretion system (T7SS) [25] can also inject toxic effectors into prey bacteria.

The large variety of T6SS toxins in *P. aeruginosa* may originate from a complex lifestyle and ability to thrive in a wide range of ecosystems. *P. aeruginosa* is a Gram-negative pathogen,

which is best known for establishing chronic and ineradicable infections in the lungs of cystic fibrosis (CF) patients [26]. After colonisation, during which *P. aeruginosa* has to outcompete the resident lung flora [27], this organism establishes a resistant and resilient biofilm that is no longer susceptible to antibiotic treatment or clearance by immune system [3]. The biofilm development involves production of an extracellular matrix consisting of exopolysaccharides, including Pel, Psl and alginate [28], while simultaneously the bacterium refrains from using motility-related devices and notably the flagellum [29]. This transition or switch in lifestyle from motile to sessile biofilms is tightly regulated by the Gac/Rsm cascade which consists of several two-component regulatory systems, small regulatory RNAs and translational repressors [30–33]. Remarkably the Gac/Rsm-dependent transition from planktonic to biofilm lifestyle [34] is accompanied by an upregulation of the 3 *P. aeruginosa* T6SSs [35,36]. This accounts for the necessity to be prepared for a fight when entering the dense polymicrobial niche where contacts with existing microorganisms will be made. The 3 *P. aeruginosa* T6SSs deliver an array of antibacterial toxins, each being produced in tandem with a cognate immunity to prevent self-intoxication [17,37]. These toxins may have a synergic impact on prey killing rather than providing redundant backup [38]. Yet the question remains whether injection of a cocktail of toxins is a robust engineering solution that guarantees that at least one toxin would be effective in a specific condition and for a specific target.

Here, we systematically assessed the importance and role of individual *P. aeruginosa* T6SS effectors using a macrocolony assay that allowed to monitor attacker and prey distribution through differential fluorescent tagging as described in previous studies [39,40]. Specifically, our prey cell lacks a single immunity which makes it sensitive to a single T6SS toxin. Our data showed that the level of T6SS activity could be upregulated in an additive manner by manipulating distinct positions in the Gac/Rsm network, and progressive increase in T6SS function is accompanied by a correlative increment in the secretion level and impact on the preys. We observed that individual toxins have variable impact on the ability of the prey to spread within the macrocolony from non-visible restriction to fully preventing any growth. We also observed synergistic effects when the role of two toxins is combined. The effectiveness of toxins also depends on the initial cell density used and the presence/absence of type IV pili as they influence the number of contacts between attacker and prey at the start and later in the competition assay. Our results are supported by biophysical simulations, that not only correctly describe how lineage distribution within community is affected by growth and interactions between individual prey and attacker bacteria, but also allow us to assess how specific parameters such as toxin dose and potency, firing rate or number of interspecies contacts determines lineage distribution within the community.

## Results

### Modulating T6SS activity level through Gac/Rsm pathway and temperature

Expression of T6SS genes in *P. aeruginosa* is partly controlled by the Gac/Rsm pathway (Fig 1A). The GacA response regulator induces expression of sRNAs, RsmY and RsmZ [41,42], which sequester two T6SS repressors, RsmA [43] and RsmN [44,45]. This cascade is negatively controlled by the sensor RetS [34,46]. Here we aimed at designing strains with various degrees of T6SS activity so that the payload of toxins delivered could be fine-tuned. We engineered T6SS transcriptional and translational reporters using promoter regions of *tssA1*, *tssA2* and *tssB3* fused to a promoterless gene encoding GFPmut3b [47]. Our data showed that in case of the H1-T6SS and H2-T6SS, both transcriptional (Fig 1B and 1C, green curve) and translational activity (S1 Fig) is elevated in Δ*rsmA* mutant strains, with disruption of *retS* further elevating T6SS expression (Fig 1B and 1C, red curve). Instead, the *rsmN* mutation has little

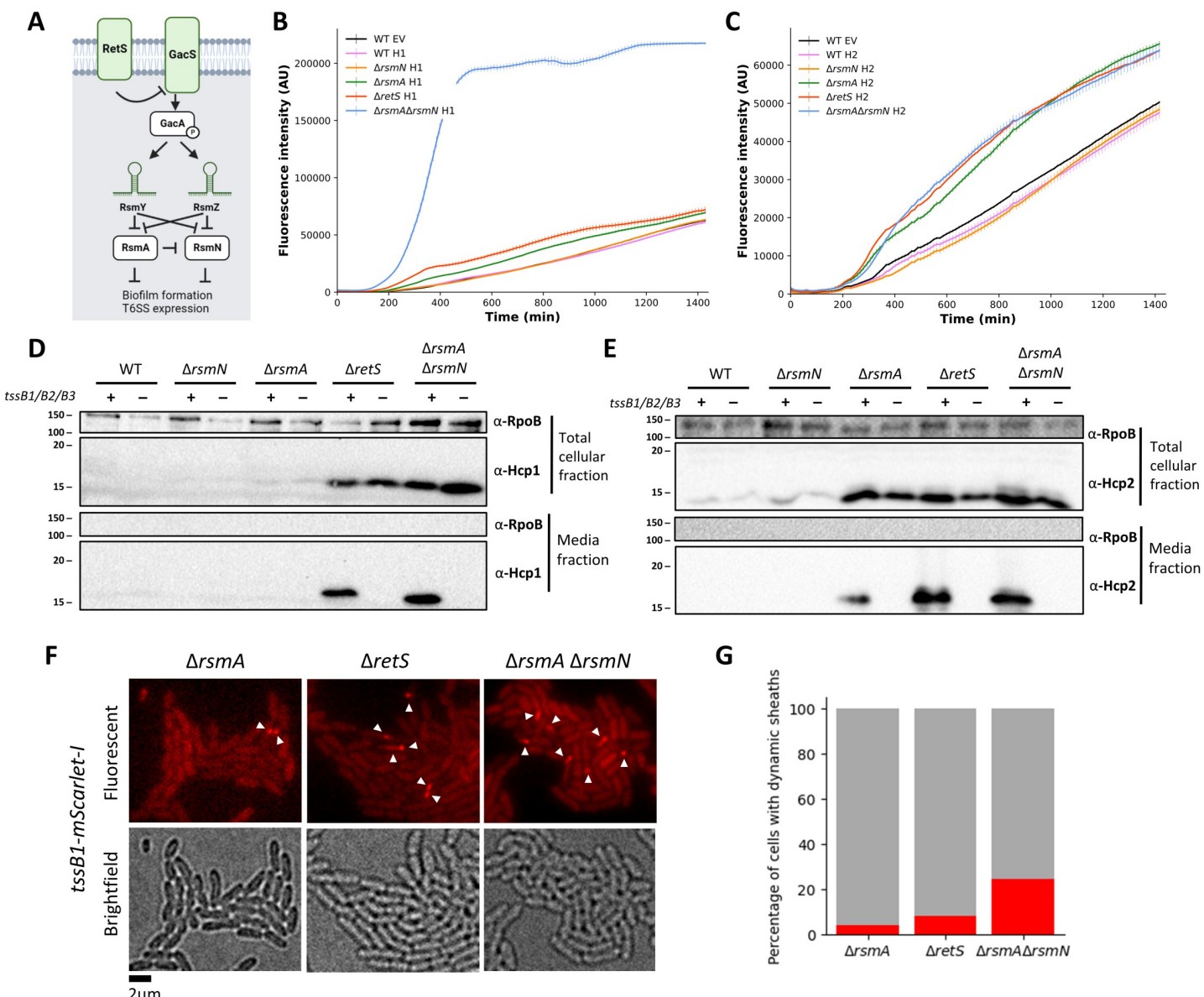

**Fig 1. Mutant strains altered in the Gac/Rsm pathway show graduated increase in both H1-T6SS and H2-T6SS expression and activity levels. (A)**. Schematic of Gac/Rsm regulatory cascade. Sensory kinase RetS represses action of GacS-GacA two component system. Upon phosphorylation of GacA, production of sRNAs RsmY and RsmZ is increased and translational repressors RsmA and RsmN are sequestered. In absence of RsmY and RsmZ, RsmA and RsmN translationally repress genes associated with T6SS and biofilm formation. (B, C) Disruption of Gac/Rsm regulatory cascade results in graduated elevation in T6SS promoter activity. Analysis of H1-T6SS (B) and H2-T6SS (C) promoter activity during growth in biofilm as measured by plasmid based GFPmut3b reporter fusions. Assessed strains include WT, Δ*rsmN*, Δ*rsmA*, Δ*retS*, and Δ*rsmA*Δ*rsmN*. All measurements are performed at 37˚C, in static culture, each display item shows a mean +SD of 4 technical replicates that is representative of n = 3 biologically independent repeats. (D, E) Western blot analysis shows gradual elevation of Hcp1 (D) and Hcp2 (E) levels in whole cell and supernatant fractions in WT, Δ*rsmN*, Δ*rsmA*, Δ*retS*, and Δ*rsmA*Δ*rsmN* strains. Presence or absence of *tssB1*, *tssB2*, and *tssB3* genes and correspondingly ability to assemble and fire T6SS sheaths is indicated by "+" or "−". A representative blot of 3 independent biological repeats is shown. Bacteria were cultured for 5h at 37˚C (H1-T6SS) and for 18h at 25˚C (H2-T6SS). RNA polymerase (RpoB) is used as lysis control. (F, G) Higher levels of H1-T6SS expression and activity are reflected by increased abundance of dynamic sheath structures. Fluorescent images illustrating changes in H1-T6SS sheath abundance in Δ*rsmA*, Δ*retS* and Δ*rsmArsmN* strains with chromosomal *tssB1*::mScarlet-I fusion at the native locus. Images are representative snapshots of S1, S2 and S3 Videos (G) Quantification of percentage of the population with dynamic H1-T6SS sheath structures over 120s timelapse. Images from 2 to 3 independent imaging session were analysed with 5000 to 6500 cells quantified for each strain shown. Bacterial cultures of $OD_{600}$ 0.05 were spotted LB agar and inverted onto cover glass. Images were taken after 3.5–4.5h incubation at 37˚C. An approx. 15.6 x 13 μm representative field of view is shown.

impact (Fig 1B and 1C, orange curve). Similar results were observed for H3-T6SS (S2 Fig). Remarkably, in the case of the H1-T6SS, the double mutation *rsmA/rsmN* drives far more expression (Fig 1B, blue curve) as compared to the single *rsmA* mutation suggesting a key role for RsmN in this case. We also observed that for both H2- and H3-T6SS, higher level of promoter activity can be observed when cells are grown at 25°C and not 37°C (S3 Fig), which agrees with previous findings [35].

We correlated T6SS expression with T6SS activity by monitoring T6SS-dependent Hcp release in the extracellular medium [48,49]. Western blot confirmed that when the T6SS is functional, the amount of Hcp1 (Fig 1D) or Hcp2 (Fig 1E) in the supernatant is incremented in an additive fashion when comparing the ΔrsmN mutant (lowest level) up to the ΔrsmAΔrsmN mutant (highest level). The secretion is not observed if the T6SS is disarmed like in a *tssB* mutant. Hcp3 production was also increased in a manner consistent with promoter activity measurements (S2 Fig). T6SS activity is also reflected in the T6SS dynamics (S1, S2 and S3 Videos), with an increase in assembly and contraction of the fluorescently-tagged T6SS sheaths [50–52]. One can clearly see that the number of dynamic H1-T6SS sheaths is gradually elevated when comparing ΔrsmA, ΔretS and ΔrsmAΔrsmN background strains (Fig 1F and 1G), which thus correlates our Hcp secretion data.

## Macrocolony set-up for imaging bacterial competition

To monitor T6SS-dependent competition, we used an assay that allows to visualise the spread and distribution of distinct *P. aeruginosa* lineages within a macrocolony. The approach is like those previously described [39,40], where distinct strains in a mix are tagged with a different fluorophore, either sfGFP or mCherry. In order to standardize the assay, we first used identical PAO1 strains, identical except for the fluorophore, which were mixed and grown on agar media resulting in microcolonies that contain 2 distinct regions. The colony central part corresponding to where bacteria are initially placed is highly mixed, and the outer colony region formed by lateral expansion contains distinct segments (Figs 2A and S4). This is reflected by increase in relative fluorescence signal variation in colony outer regions and corresponding quantifiable reduction in strain intermixing (S5 Fig). Colony structures and the extent of strain mixing in absence of competition is determined by bacterial density in the initial inoculum. Communities appear gradually less mixed as the inoculum density decreases from $OD_{600}$ 1 to $OD_{600}$ 0.001 (Fig 2B). Additionally, colony morphology is subject to disruption of the Gac/Rsm pathway, as both ΔretS and ΔrsmAΔrsmN strains form smaller and more aggregative colonies.

Next we used this assay to test whether the ability to fire T6SS provides an advantage in otherwise isogenic competition. A triple deletion of *tssB1*, *tssB2*, and *tssB3* rendering all T6SSs non-functional, was engineered in a ΔretS background where system expression levels are high. The competition between the triple T6SS mutant used as a prey and the parental ΔretS strain used as an attacker did not show any advantages beyond inherent structure variability (S6 Fig). This suggests that the presence of all T6SS immunities in the prey cell is sufficient to resist any T6SS-dependent killing. Considering these results, we used this assay to investigate the action of individual T6SS toxins.

## The impact of individual toxins on sensitized preys shows a high degree of variability

To assess the impact of an individual T6SS toxin, it is essential to have a prey and attacker which are isogenic and differs only by the absence of a single T6SS immunity in the prey. We systematically engineered *P. aeruginosa* strains that lack an immunity protecting from either

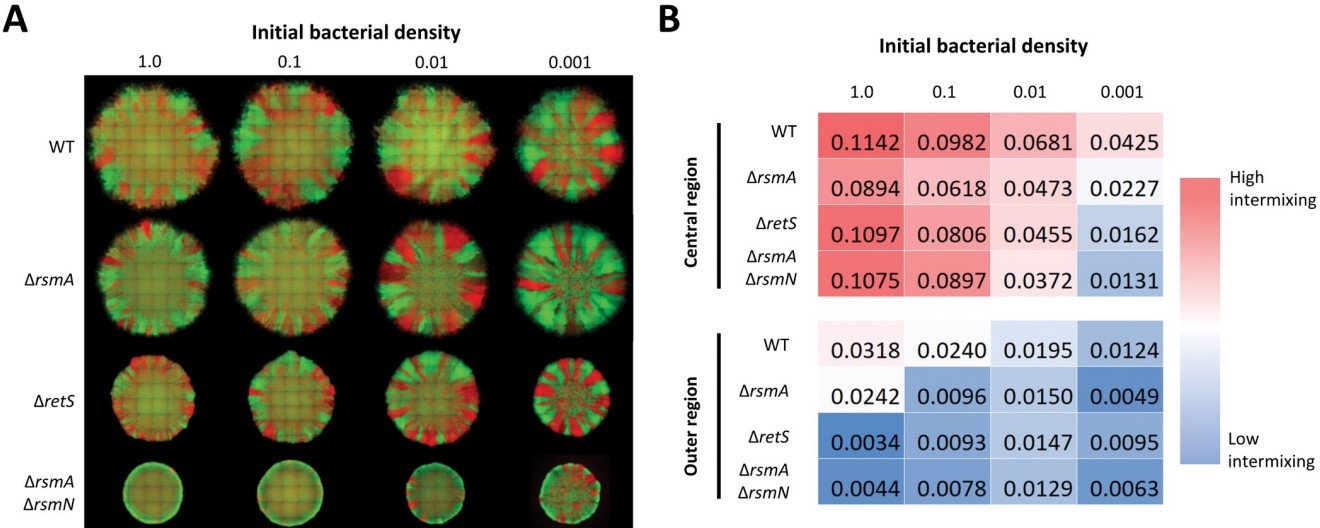

**Fig 2. Colony morphology is determined by strain background and bacterial density in inoculum.** (A) Set of representative fluorescence composite images of mixed bacterial colonies of WT, Δ*rsmA*, Δ*retS*, and Δ*rsmA*Δ*rsmN* strains with altered inoculum densities. Isogenic bacterial strains tagged with mCherry (red) and sfGFP (green) fluorophores were mixed at 1 to 1 ratio and after adjusting inoculum density (OD$_{600}$ 1.0; 0.1; 0.01; 0.001) spotted on LB agar, images of whole macrocolonies taken after 48h incubation at 37˚C show 2 morphologically distinct regions—highly mixed inner region corresponding to inoculum zone and outer region where spatial segregation of the sub-populations is apparent. A representative image of 3 biological replicates is shown, corresponding individual fluorescence channel images are shown in S4 Fig. (B) Mean sub-population intermixing changes in colony central and outer regions for the corresponding colonies. Spatially resolved mean channel fluorescence and strain intermixing analysis is shown in S5 Fig.

one of all known H1- and H2-T6SS toxins (Table 1) [38,53,54]. To avoid self-intoxication of the mutant prey strain upon deletion of the immunity gene, the genes encoding both the immunity and the cognate toxin were removed. The deletions were introduced in the wild type (WT) and T6SS active strains with disruptions in the Gac/Rsm cascade. For purposes of performing the macrocolony assay, toxin-sensitised strains, or "preys" were tagged with constitutively expressed mCherry, whereas the "attacker" strain, with or without a functional T6SS, was tagged with sfGFP.

When the attacker strain is the wild type, *i.e.* no mutation in the Gac/Rsm pathway, "prey" immunity mutants were barely restricted in growth, as could be seen by "prey" expansion when looking at the red channel (S7A Fig). In the Δ*rsmA* genetic background, where the T6SS activity is increased, one can observe a restricted expansion in the macrocolony, as seen by the decreased coverage of red fluorescence, for several strains sensitized to H1-T6SS toxins (Fig 3A). Here we quantified these differences by looking at a set of 3 biological replicate macrocolony communities and perform analysis of fluorescence signal spatial distribution (S8 Fig). Looking at relative fluorescence signal mean intensity as a function of distance from the colony centre, we confirm that in the given macrocolonies, bacteria appear equally distributed across the whole of the inoculum region independently of the attacker populations T6SS activity or toxin used in the competition. However, a stark drop in mean prey signal intensity can be seen in competition reliant on delivery of toxins Tse2, Tse4, Tse5, Tse6, and Tse7 (S8 and S9 Figs). From these observations one can see that among the most potent H1-T6SS toxins are Tse4 [38] and Tse5, previously called RhsP1 [61], which are both pore-forming toxins, together with the NAD-dependent Tse2 toxin [58] (Fig 3A). It is also to be noted that increasing further the H1-T6SS activity using Δ*retS* or Δ*rsmA*Δ*rsmN* mutants, drastically improve the impact of other toxins such as Tse3, Tse6 and Tse7 (Figs 3B and S7B). For strains sensitized to H2-T6SS toxins (S10 Fig), several "preys" no longer expand but such clear difference is seen at 25˚C and not at

**Table 1. *Pseudomonas aeruginosa* T6SS toxins and effectors.**

| | Effector | Cognate toxin immunity | Description | References |
|---|---|---|---|---|
| **H1-T6SS** | Tse1 (PA1844) | Tsi1 (PA1845) | Amidase that hydrolyses peptidoglycan crosslinks causing target cell lysis | [55,56] |
| | Tse2 (PA2702) | Tsi2 (PA2703) | NAD dependent ADP-ribosyltransferase that causes quiescence, toxic when expressed in eukaryotic cells | [55,57,58] |
| | Tse3 (PA3484) | Tsi3 (PA3485) | Muramidase that hydrolyses peptidoglycan backbone causing target cell lysis | [55,56] |
| | Tse4 (PA2774) | Tsi4 (PA2775) | Forms ion selective membrane pores that disrupt proton motive force | [38,59] |
| | Tse5 /RhsP1 (PA2684) | Tsi5 /RhsP1i (PA2683.1) | Forms ion selective membrane pores resulting in loss of membrane potential, contains RHS and PAAR domains | [59–61] |
| | Tse6 (PA0093) | Tsi6 (PA0094) | NAD(P)+ glycohydrolase that depletes NAD(P)+ resulting in target bacteriostasis, contains PAAR domain | [59,61,62] |
| | Tse6$^{PA14}$ /Tas1 (PA14_01140)* | Tis1 (PA14_01130) * | (p)ppApp synthetase that rapidly depletes prey ATP/ADP pool, contains RHS and PAAR domains | [63] |
| | Tse7 (PA0099) | Tsi7 (PA0100) | Tox-GHH2 nuclease, contains PAAR domain | [54,61] |
| | Tse8 (PA4163) | Tsi8 (PA4164) | Targets transamidosome complex disrupting protein synthesis | [53,64] |
| **H2-T6SS** | Tle1 (PA3290) | Tli1a (PA3291) Tli1b (PA3292) | Phospholipase A2 activity | [65–67] |
| | Tle2 (PAK_03765)* | Tli2a (PAK_03766) * Tli2b (PAK_03767) * | Phospholipase A1 activity | [67] |
| | Tle3 (PA0260) | Tli3 (PA0259) | Phospholipase | [67,68] |
| | Tle4 /TplE (PA1510) | Tli4 /TplEi (PA1509) | Phospholipase, contains PGAP1-like domain to aid internalisation into host cells and localisation at ER membrane inducing autophagy, both antibacterial and anti-host toxicity | [67,69,70] |
| | PldA /Tle5a (PA3487) | Tli5a (PA3488) | Phospholipase D with preference for PE (phosphatidylethanolamine) as substrate, activate PI3K/Akt pathway that promotes internalisation into host cells, both antibacterial and anti-host toxicity | [67,69,71–74] |
| | PldB /Tle5b (PA5089) | Tli5b$_1$ (PA5088) Tli5b$_2$ (PA5087) Tli5b$_3$ (PA5086) | Phospholipase D, activates PI3K/Akt Pathway that promotes internalisation into host cells, both antibacterial and anti-host toxicity | [67,69,73,75] |
| | VgrG2b (PA0262) | VgrG2bi (PA0261) | Contains zinc- metallopeptidase domain, hydrolyses peptidoglycan in cell division plane causing target lysis. Interacts with α- and β-tubulins and γTuRC promoting internalisation into nonphagocytic cells, both antibacterial and anti-host toxicity | [68,76,77] |
| | AmpDh3 (PA0807) | AmpDh3i (PA0808) | Cell wall remodelling amidase that contributes to β-lactam resistance, hydrolyses target cell peptidoglycan cross links | [78–80] |
| | TseT (PA3907) | TsiT (PA3908) | TOX-REase domain containing nuclease | [81] |
| | TseV (PA0822) | TseV (PA0821) | Putative VRR-NUC domain containing nuclease | [82] |
| | PA5265 | PA5264 | Contains MIX (marker for type six effectors) motif | [83] |
| | RhsP2 (PA14_43100) * | RhsP2i (PA14_43090)* | ADP-rebosyltransferase that targets cells ncRNA pool causing growth arrest, contains RHS domain | [61,84,85] |
| | PA2066 | Unknown | Putative Hcp2 associated effector | [48] |
| | Azu (PA4922) | - | Common good effector, copper binding and uptake | [86,87] |
| | ModA (PA1863) | - | Common good effector, molybdite binding and uptake | [82,88] |
| **H3-T6SS** | PA0256 | Unknown | Putative Hcp3 associated effector | [48] |
| | TseF (PA2374) | - | Common good effector, interacts with PQS and the siderophore receptor FptA aiding in iron uptake | [89] |
| | TepB (PA14_33970)* | | Common good effector associated with improved abiotic stress tolerance under biofilm condition, translocated to host cells | [90] |

*Not conserved in *P. aeruginosa* PAO1

37˚C in agreement with our T6SS expression data (S3 Fig). As for the H2-T6SS, our data showed that the two most potent toxins are TseT [81], a nuclease, and PA5265 [91], an effector of unknown function that carries a MIX motif [83] (S10 Fig).

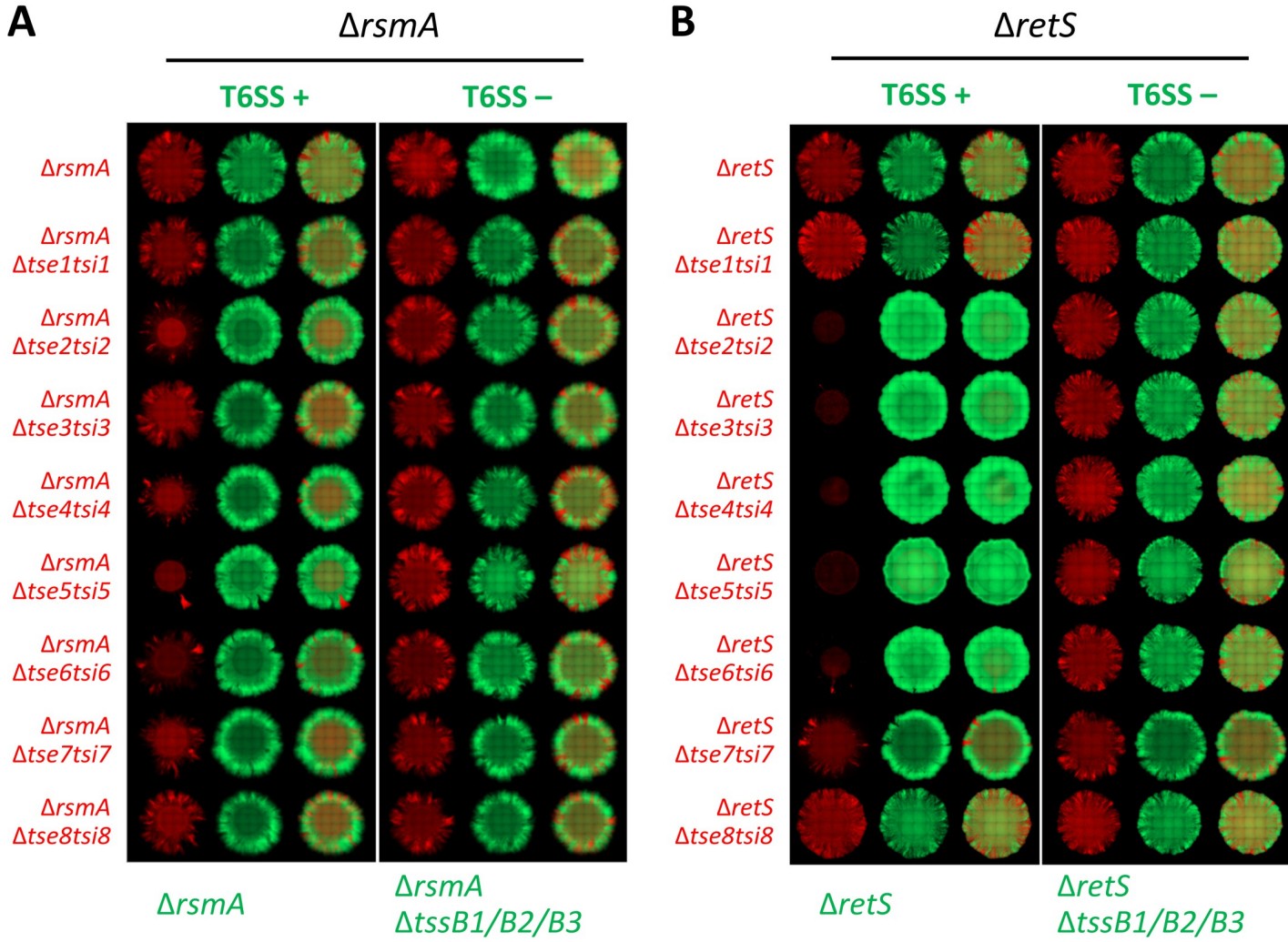

**Fig 3. Toxin sensitive prey growth restriction is dependent on the type of toxin and dose delivered. (A, B)** Representative image of 48h old mixed colonies of toxin sensitised bacteria in red in competition with T6SS+ or T6SS- (ΔtssB1ΔtssB2ΔtssB3) strains of the same regulatory background in green. Upper lane contains a control mix of bacteria with full toxin-immunity gene sets, each of the following lanes contains strain sensitised to one of the H1-T6SS toxins from Tse1 to Tse8. Images sets of competitions of ΔrsmA (**A**) and ΔretS (**B**) regulatory background strains shown with each of the sets containing both single fluorescence channel and overlay images showing distribution of sensitised prey in a mix with T6SS⁺ and subsequently T6SS- (ΔtssB1ΔtssB2ΔtssB3) parental strain. Strains contain constitutively expressed fluorescent proteins, prey labelled with mCherry (shown in red) and attacker with sfGFP (shown in green). All bacteria mixed at 1:1 ratio, inoculum $OD_{600}$ = 1.0, grown for 48h at 37˚C on LB with 2% (w/v) agar.

In order to ascertain that the phenotypes observed are mostly due to the sensitization of a prey strain to a given toxin, we also performed complementation experiments by reintroducing in trans the cognate immunity gene. We chose the strains sensitized for Tse2 and Tse5 (H1-T6SS) as well as TseT and Tle3 (H2-T6SS). The corresponding immunity genes were cloned into pBBRMCS-5 and introduced in the appropriate mutants. For this assay we used a strain with a ΔretS background for assessing H1-T6SS and ΔrsmA for assessing H2-T6SS. The microcolony assay data are shown in S11 Fig. and showed a clear restoration of the ability of the prey strain, now expressing the immunity, to cope with the attacker, and being able to expand (red channel) in the microcolony. Whereas the complementation is obvious for Tse2, TseT and Tle3 toxicity, it is far more subtle for Tse5. This barely detectable complementation

could be due to several factors, such as the low copy number of immunity proteins as compared to the toxins and/or the high toxicity of Tse5.

## Synergy and injection loads are key modulators of T6SS toxin effectiveness

The observed discrepancy in T6SS toxin impact was investigated further to identify parameters which may fine tune our observations and to find conditions which will unveil the role of additional toxins. One such parameter could be that some toxins may act better in synergy rather than individually, which was previously proposed [38]. Interestingly both Tse1 and Tse3 impact peptidoglycan biogenesis [56], either by cleaving the peptide bonds or sugar backbone, respectively. Although Tse1 and Tse3 have little impact on their own (Fig 3A), when a strain was sensitised to both toxins, its growth was now spatially restricted when in competition with a T6SS active strain (Fig 4A), demonstrating an additive effect between the two T6SS toxins.

Another parameter to play with to unveil toxin impact is to vary the injection load. Here we used toxins of theoretically different potency, such as Tse1, Tse3, Tse2 and Tse5, from mildest ones (Tse1 and Tse3) to more potent (Tse2 and Tse5). We also used well-designed genetic background, including WT or Δ*rsmA*, Δ*retS* or Δ*rsmA*Δ*rsmN* mutant, from the lower T6SS activity to the highest, respectively. In this context we clearly detect a gradually differentiable change in competitive outcome. For example, sensitisation to Tse3 is now readily detectable in Δ*retS* or Δ*rsmA*Δ*rsmN* mutants (Fig 4B), which suggests a correlation between high T6SS activity and higher toxin payload in the prey. Our data also flagged that Tse5 is more potent than Tse2, whereas sensitisation to Tse1 is not effective in any of the backgrounds (Fig 4B). This may indicate that altering the integrity of the peptidoglycan backbone has a more severe impact on cell integrity as compared to compromising the bond between the glycan strands. Increasing the payload also improve some of H2-T6SS toxin impact, as shown for phospholipases PldA, and to a lesser extent Tle1, that did not appear to provide notable competitive advantage in Δ*rsmA* strains (S10 Fig) but were now effective in a Δ*retS* background (S12 Fig). This emphasizes that increase in T6SS activity leads to a competitive advantage and suggests that not only the activity of the toxin is important, but the amount delivered is a key parameter for toxin with lower potency.

## Inoculum composition determines the number of prey/attacker contacts within macrocolony and subsequent survival under T6SS attack

Our experiments provided a systematic exploration of the T6SS-mediated interactions, such as toxin potency or firing efficacy affecting community structure. To explore how local population intermixing within bacterial microcolonies impacts T6SS-dependent competition, we complement the experimental work with theoretical simulations using the platform CellModeller (Fig 5) [30]. The outcome of competition is hypothetically dependent on direct cell-to-cell contact being established between the competitors, as this maximises chances of T6SS injection into susceptible prey bacteria. The number of contacts between two competitors can be manipulated in several ways including by changing the proportion of "prey to attacker" strains or by a global density increase in the inoculum mix (S13 and S14 Figs).

We thus used a set of simulations reflecting variation in these parameters in starting population (S13 and S14 Figs). Changes in inoculum density or prey/attacker ratio have been shown to affect whole colony morphology [92,93] and to change outcomes of non-T6SS based bacterial competition [39,40]. We showed here that changes in inoculum density and mixing ratio affect population time resolved dynamics in a different way (S14 Fig), but both will influence the level of local population intermixing and number of interspecies contacts upon dense community establishment (S13B Fig). Low initial bacterial density allows for sub-population establishment before a dense community is formed, this results in spatially segregated

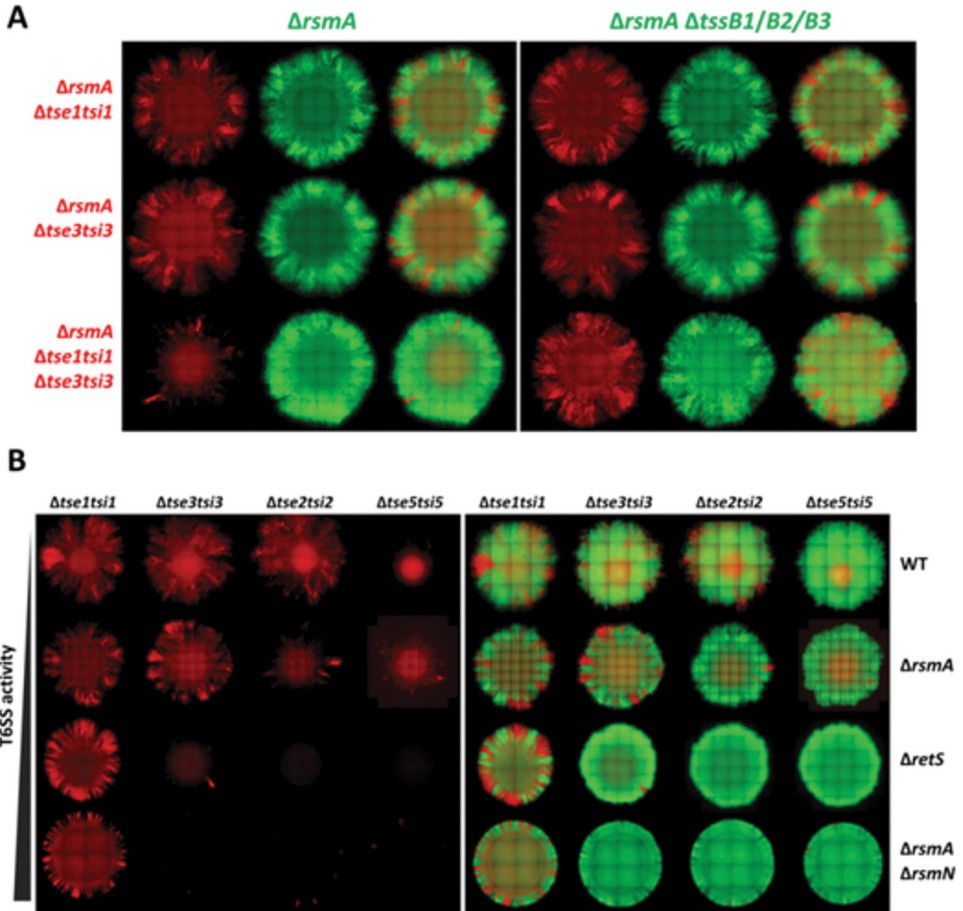

**Fig 4. T6SS toxin potency differs and increases with elevation in T6SS activity or synergistic toxin action. (A)** Representative fluorescent images of Δ*rsmA* strains sensitised to Tse1 or Tse3 individually and in combination. Both single fluorescence channel and overlay images showing distribution of sensitised prey in a mix with T6SS+ and subsequently T6SS- (Δ*tssB1*Δ*tssB2*Δ*tssB3*) parental strain. Strains contain constitutively expressed fluorescent proteins, prey labelled with mCherry (shown in red) and attacker with sfGFP (shown in green). All bacteria mixed at 1:1 ratio, inoculum OD$_{600}$ = 1.0, grown for 48h at 37˚C on LB with 2% (w/v) agar. **(B)** Whole colony fluorescent microscopy image of T6SS sensitive strain distribution within a colony when mixed with a T6SS+ strain. Corresponding fluorescent channel composite images showing toxin sensitive bacteria in red and T6SS+ parental strain in green. Strains sensitised to individual H1-T6SS toxins in each of the columns in the following order: Tse1, Tse3, Tse2, and Tse5. Each row contains strains with different regulatory background in following order of ascending H1-T6SS activity: WT, Δ*rsmA*, Δ*retS*, and Δ*rsmA*Δ*rsmN*. Strains contain constitutively expressed fluorescent proteins, prey labelled with mCherry (shown in red) and attacker with sfGFP (shown in green). All bacteria mixed at 1:1 ratio, inoculum OD$_{600}$ = 1.0, grown for 48h at 37˚C on LB with 2% (w/v) agar.

monoclonal clusters as reflected by the decrease in interspecies contact assortment (S14 Fig). Theoretical modelling shows that decrease in initial cell density results in gradually increased prey survival (Fig 5C) with the extent of the competition effectiveness remaining proportional to changes in the attacker T6SS firing rate and the spatial distribution of remaining prey being determined by the toxin lysis rate (S15A Fig).

To draw a direct comparison between predictions and experimental data we supplemented simulations using our macrocolony set-up with prey strains lacking immunity for Tse5, the most potent of the H1-T6SS toxins (Fig 5). We inoculated plates and started the growth of the macrocolony using initial OD$_{600}$ ranging from 0.001 to 1 (Figs 5D, S15B and S15C). We observed that at the highest initial OD$_{600}$, *i. e.* 1, the prey sensitized against Tse5 does not

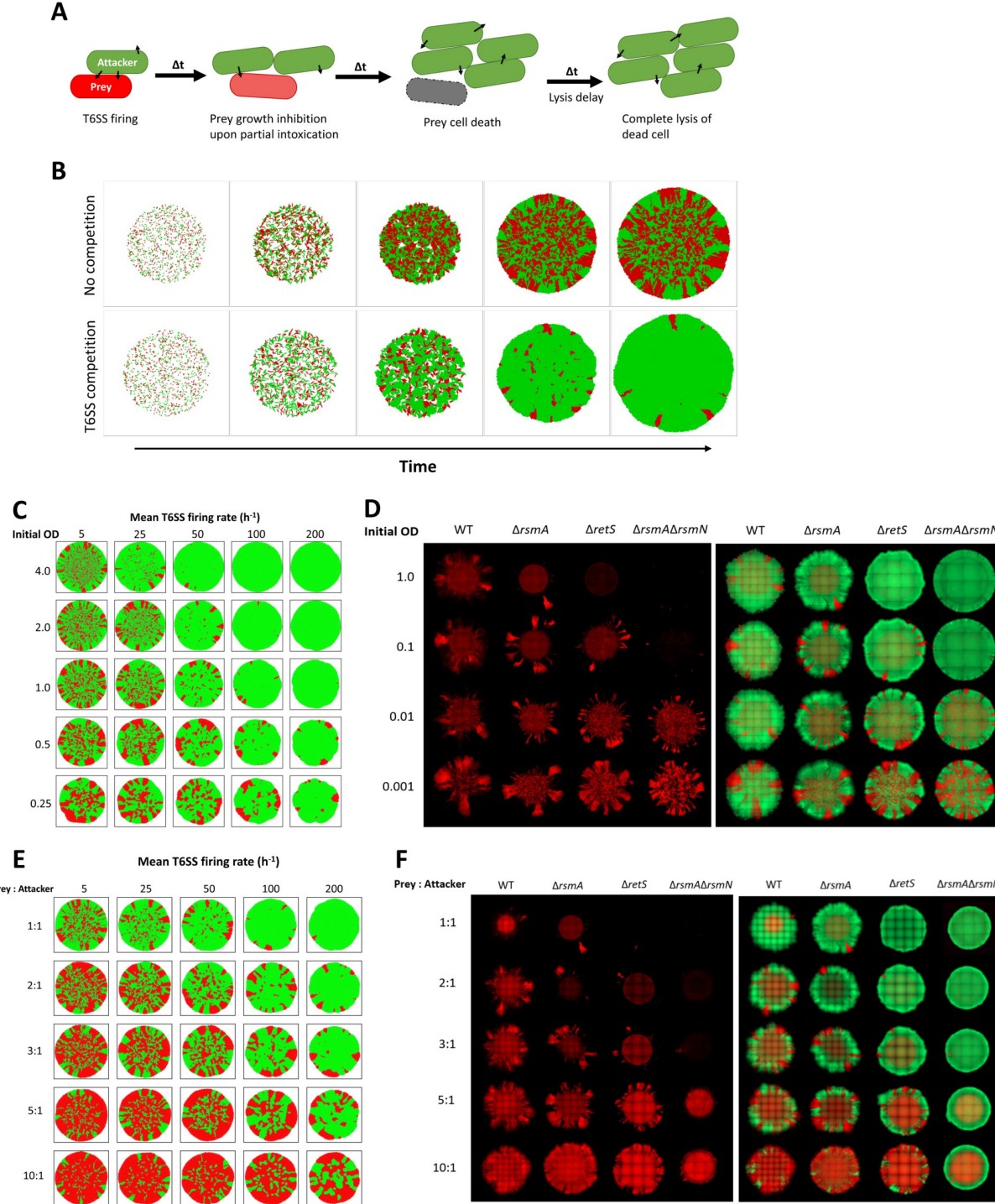

**Fig 5. Prey establishment and decrease in inter-population contacts protects T6SS sensitive lineages. (A)** Schematic depicting types of bacteria represented within the model and their fate upon T6SS-based interactions. T6SS+ attacker in green can fire a specified number of T6SSs in random directions and grows over time. Prey bacteria in red are susceptible to T6SS attacks, when prey is hit by T6SS its growth rate is reduced. Once certain threshold of T6SS attacks is reached prey cell dies (grey) and no further growth occurs. Dead prey cells remain in simulation until lysis defined by a certain delay is reached and are subsequently removed. **(B)** Simulation of 2 mixed colony development over time with and without T6SS mediated competition. Initially 1 to 1 mix of bacteria from both populations are randomly scattered on surface in circular inoculum zone. Bacterial growth occurs through elongation and division and in absence of active movement bacteria expand through rigid body interactions. At a whole colony scale, inoculum zone is filled by intermixed clonal clusters and further expansion is restricted by neighbouring cells exerting forces upon each other. Over simulation time-course radial expansion results in formation of distinct sectors in the

region outside the inoculum area. In a mix with T6SS interactions, sensitive prey in red can be seen to initially establish in the colony centre, but over simulation time-course be gradually eliminated by T6SS$^+$ attacker in green. Specific parameter space of simulations is shown in S4 Table. Representative simulation outputs showing how decrease in initial bacterial density (**C**) or changes in strain mixing ratio (**E**) within initial population promotes prey (red) survival in a mix attacker population (green) with differing mean of T6SS firing rate and within a context of highly lytic toxins. (Toxin lethal dose = 5, lysis delay = 3min). Tse5 mediated competition outcomes in context of strains with differing T6SS activity levels using populations inoculated from different initial density (**D**) or changing the prey to attacker ratio within inoculum (**F**). Representative image of 48h old mixed colonies of Tse5 sensitised bacteria in red in competition with T6SS+ strains of the same regulatory background in green. Mixed bacterial colonies of WT, Δ*rsmA*, Δ*retS*, and Δ*rsmA*Δ*rsmN* strains in different columns. First set of images shows prey distribution only and second set shows corresponding fluorescent channel overlay of both prey and attacker populations. Strains contain constitutively expressed fluorescent proteins, prey labelled with mCherry (shown in red) and attacker with sfGFP (shown in green). Isogenic bacterial strains tagged with mCherry (red) and sfGFP (green) fluorophores were mixed at 1 to 1 ratio and after adjusting inoculum density (OD$_{600}$ 1.0; 0.1; 0.01; 0.001) spotted on LB agar to test impact of inoculum density (**D**). In (**F**) each of the rows contains a set of representative images of colonies set up with a different prey to attacker ratio in the inoculum (1:1, 2:1, 3:1, 5:1, 10:1) and the overall bacterial density of OD$_{600}$ = 1.0. Images of whole microcolonies taken after 48h incubation at 37°C on LB with (2% w/v) agar, a representative image of 3 biological repeats is shown.

expand in Δ*retS* and Δ*rsmA*Δ*rsmN* mixes, with small number of bacteria growing in the outer colony region in WT and *rsmA* strains. Instead, at the lowest densities, *i. e.* 0.001, prey spreads nearly as much as the attacker even in Δ*retS* and Δ*rsmA*Δ*rsmN* mixes (Fig 5D). This agrees with our experimental observations shown in Fig 2, that at low initial density, two strains within a macrocolony can form distinctive zones whereas at higher density there is no zone formation but a uniform mix (Fig 2A and 2B), suggesting that in the former case prey clonal expansion is allowed likely due to fewer contacts between populations that result in fewer opportunities to deliver T6SS toxins.

Prey-attacker mixing can also be adjusted by changing proportion of strains in the inoculum. A 1:1 population mixing ratio results in highest contact assortment while decrease in interspecies contact formation is seen as population mixing ratio departs from 1:1 (S13 and S14 Figs). Our simulations show indeed that increasing the amount of prey in inoculum favours its survival, and further departure from 1:1 mixing is required to protect sensitive prey in cases where T6SS attack frequency is elevated (Fig 5E). This can also be supported experimentally using Tse5-dependent competition (Figs 5F and S16). When the ratio is 1:1 the Tse5-sensitive prey is readily outcompeted in all genetic backgrounds. However, the attacker advantage is progressively lost when increasing the prey ratio. Even in case of competition in Δ*rsmA*Δ*rsmN* genetic background, where the prey is almost eliminated if bacteria are mixed at 1:1 ratio, large amounts of prey bacteria are present at the inoculum region when mixed at 5:1 or 10:1 ratio.

Combining theoretical and experimental approaches we confirmed that T6SS-mediated killing is dependent not only on the range of toxins delivered and the toxin dose, but also "attacker" population being able to establish direct contact with prey cells.

## Type IV-pili-dependent intermixing and motility are key players in controlling T6SS-mediated killing

Gac/Rsm cascade controls not only *P. aeruginosa* T6SS activity but is the central regulatory cascade in planktonic to biofilm lifestyle switch involving hundreds of genes [33,43]. Among RsmA/RsmN regulatory targets are genes associated with motility, including genes encoding type IV pili (T4P) components [43]. T4P are used for cell-cell interaction, cell-surface attachment, and surface "twitching motility" [94]. In phenotypic analysis comparing the five Gac/Rsm cascade mutant strains in T4P-mediated motility assay, Δ*rsmA*, Δ*retS* and Δ*rsmA*Δ*rsmN* mutants all showed a significant impairment in twitching (Fig 6A and 6B), with the Δ*rsmA*rsmN* mutant specifically showing more than 90% reduction in twitching motility relative to WT. This is consistent with observation that both Δ*retS* and Δ*rsmA*Δ*rsmN* strains form smaller aggregative colonies with measurably less intermixed subpopulations (Figs 2, S4 and S5).

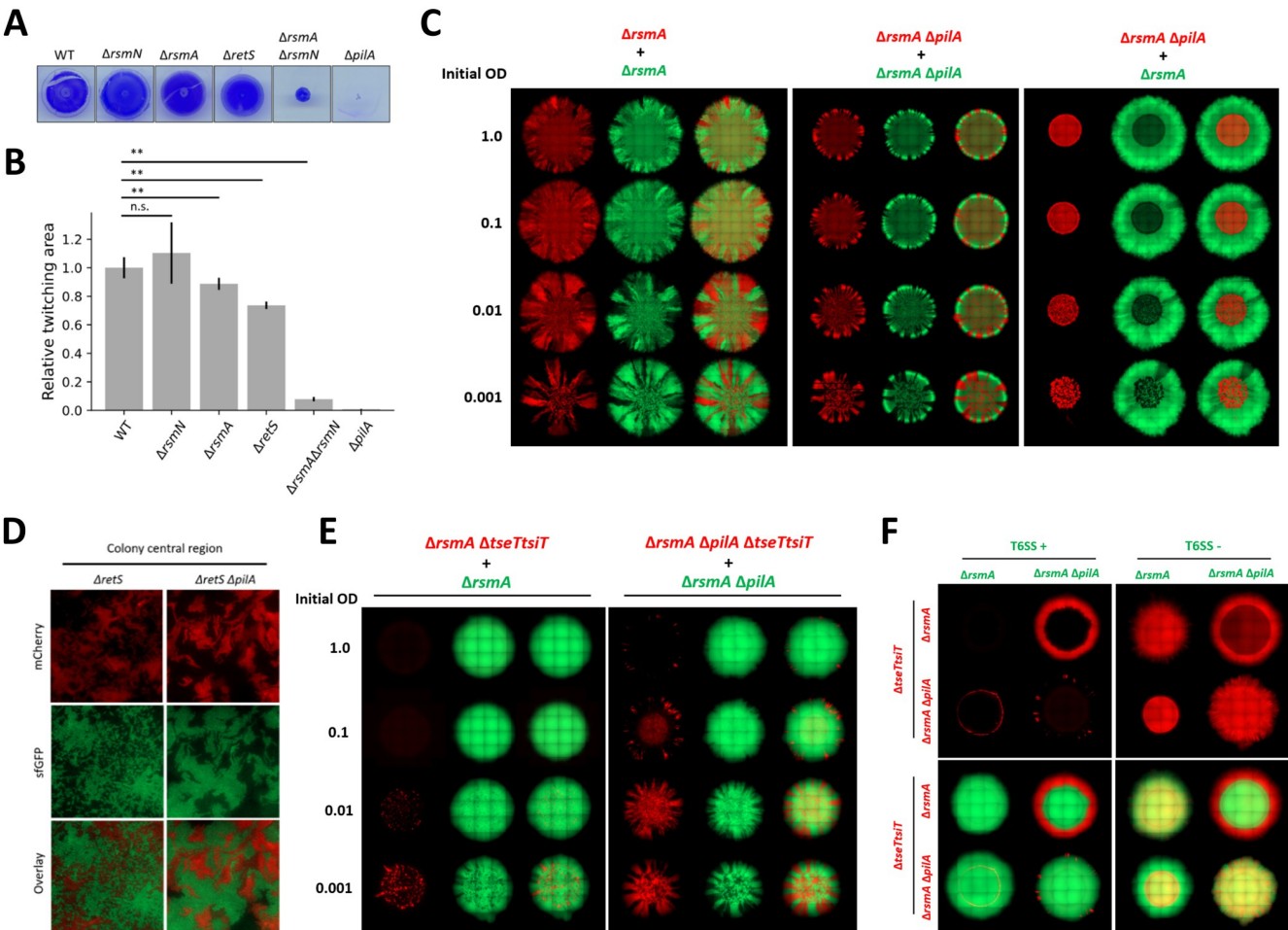

**Fig 6. Loss of T4P hinders *P. aeruginosa* ability to intermix within communities resulting in impaired T6SS killing. (A, B)** Twitching assay assessing surface motility of mutants in the Gac/Rsm pathway and Δ*pilA* mutant strains as compared to WT. Images comparing strain twitching zones as seen by crystal violet staining (**A**) and corresponding quantification of the relative twitching zone area (**B**). (Mean + SD of n = 5 biological repeats). **(C)** A set of mixed bacterial macrocolonies consisting of T4P+ (Δ*rsmA*) and T4P- (Δ*rsmA*Δ*pilA*) strains tagged with constitutive mCherry (red) and sfGFP (green). Single fluorescence channel and overlay images of T4P+ and T4P+, T4P- and T4P-, T4P- and T4P+ mixed colonies, displayed to scale. **(D)** Bacterial T4P+ (Δ*retS*) and T4P- (Δ*retS*Δ*pilA*) mutant intermixing at single-cell scale. Mix of isogenic strains with mCherry and sfGFP tags grown in the interface between media agar and glass coverslip until dense community is established. Single fluorescence channel and overlay images show the different extent of local intermixing. **(E)** A set of mixed bacterial communities where TseT sensitive prey growth is spatially restricted in presence of T6SS+ (Δ*rsmA*) parental strain. Showing both individual fluorescent channel and overlay images of both prey population in red and attacker in green. Toxin sensitised strain mixed with a parental strain at 1 to 1 ratio and after adjusting inoculum density (OD$_{600}$ 1.0; 0.1; 0.01; 0.001) spotted on LB agar to assess how reduction in lineage intermixing impacts T6SS mediated competition. **(F)** TseT-mediated competition was used to assess impact on T6SS mediated killing in a mix of T4P+ and T4P- bacteria. The panel contains individual red channel images showing toxin sensitive prey distribution in presence of T6SS+ (left) or T6SS- (right) attacker strain in the upper two rows. Corresponding overlay images showing distribution of both toxin sensitive prey (in red—mCherry) and T6SS attacker population in green (sfGFP) shown in the lower two rows. T4P+ prey (Δ*rsmA*Δ*tseTtsiT*) strains in the upper row, with T4P- prey (Δ*rsmA*Δ*pilA*Δ*tseTtsiT*) in the lower row. With T4P+ attacker in the first column and T4P- attacker strains in the second. All bacteria mixed at 1 to 1 ratio, with inoculum density as specified for **(C)** and **(E)**, or OD$_{600}$ 1.0 for **(F)**, grown for 48h at 37°C on LB with 2% (w/v) agar for **(C)** and grown at 25°C on LB with 1.2% (w/v) agar for **(E, F)**. 1 of 1 biological repeat shown for all images containing Δ*pilA* mutant strains.

Within our macrocolony context, *P. aeruginosa* expansion on the agar rigid surface is associated with T4P-mediated motility. When comparing *P. aeruginosa* T4P+ and T4P- strain (Δ*pilA* mutant) whole colony morphology, it is notable that similarly to Δ*rsmA*Δ*rsmN* mutant, T4P- colony outer region is reduced in size and strains are less intermixed (Fig 6C). Additionally, loss of T4P results in reduced intermixing at single cell scale (Fig 6D), with the Δ*pilA* mutants present in monoclonal clusters. This reduction in local intermixing and formation of

monoclonal clusters is comparable to the theoretical sub-population segregation observed in cases of population establishment at low inoculum density (Fig 5). To test how loss of T4P and the associated reduction in intermixing affects T6SS-associated killing we used the macrocolony competition assay mediated through the H2-T6SS toxin TseT and inoculated bacterial colonies with mixes of different densities to allow direct comparison of prey killing efficacy (Fig 6E). In case of T4P+ strains, growth of TseT-sensitive bacteria was entirely restricted in colonies with inoculum $OD_{600}$ of 1.0 or 0.1, while at lower densities small amounts of prey were present in the colony central region. In contrast, in competition between T4P- prey and attacker strains, TseT sensitised bacteria were detectable in colonies with higher inoculum density. This indicates that ability to actively intermix using T4P is contributing to *P. aeruginosa* ability to eliminate toxin sensitive prey.

Loss of T4P is deleterious to *P. aeruginosa* ability to eliminate T6SS-sensitive prey likely reflecting a drastic reduction in T4P-dependent cell-cell contact (Fig 6E). Since strains lacking *pilA* form smaller colonies (Fig 6C), and are impaired in twitching motility, we therefore assessed whether a motile prey could somehow run away from a non-motile attacker (Δ*pilA* mutant). Firstly, in mixed colonies containing both T4P+ and T4P- strains, bacteria lacking T4P are present in the inoculum region, the laterally expanding colony edge is exclusively occupied by T4P+ strains and the colony overall size is comparable with the T4P+ colonies (Fig 6C). This sub-population distribution is consistent with the observation that T4P are likely the primary *P. aeruginosa* mode of active expansion within the context of macrocolony assay and only the motile (T4P+) population can expand in the outer colony regions. Subsequently, in a mix where a non-motile T4P- prey (Δ*rsmA*Δ*tseT*Δ*tsiT*Δ*pilA*) is in presence of T6SS+ attacker, sensitive bacteria are almost eliminated (Fig 6F). The exception is that some of the prey bacteria appears to survive at the very edge of the inoculum spot, which might be due to a microfluidics phenomenon termed "coffee ring effect" [95]. A possible explanation for prey survival here being the T4P+ cell inability to intermix with a dense T4P- population and subsequently failing to eliminate the sensitive bacteria. Remarkably, in a mix where motile T4P+ prey (Δ*rsmA*Δ*tseT*Δ*tsiT*) bacteria are in presence of non-motile T4P- attacker, the prey bacteria are eliminated from the colony central region, but able to expand and grow in the colony outer edge to the same extent as in absence of T6SS-mediated attack. This T4P-mediated impact on T6SS dependent killing in *P. aeruginosa* also extends to H1-T6SS-sensitized prey, like a strain lacking Tse5 immunity (S17 Fig), thus supporting that the effect observed is likely universal rather than specific to the strain or even the type of contact based killing system used.

In summary, we showed that T4P promoting cell contact and surface motility determines the outcome of contact-mediated competition. We have further shown that the differences observed are not due to any changes in the Δ*pilA* mutant T6SS activity as compared to the isogenic parental strain (S18 Fig) comparing Δ*rsmA*Δ*pilA* and Δ*rsmA*. T4P appears to minimise interspecies contacts and promote escape from T6SS attacks through segregation, with two underlying mechanisms being—prey "running" from a non-motile attacker or prey escaping through loss of T4P as the attacker cells are no longer able to "hold onto" susceptible prey.

## Discussion

Bacterial competition or warfare is of huge importance for the survival and prevalence of species within complex populations, such as ones found in the plant rhizosphere [96] or the gut microbiota [97,98]. Understanding and predicting the fate of polymicrobial populations is a key question in microbiology since for example reprogramming microbiota will help fight obesity, colonic diseases or simply prevent pathogen invasion [99]. Understanding the rumen

microbial organisation would have industrial application for the degradation of plant cell walls [100]. Capturing appropriate soil microbiome composition would support specific agricultural needs [101] or even aid in targeted forest restoration [102]. One of the challenges in the field is engineering of relevant experimental model systems that can capture polymicrobial interactions. Studies aiming to understand dynamics of polymicrobial populations substantially benefit from integrating both experimental and theoretical models as those presented here.

An equilibrium within a microbial population can be reached when mutual benefits prevail or upon selfish expansion of a single species [103]. Benefits include ability of some species to catabolise specific nutrient sources yielding products which can be used by others. It also involves the ability of some species to protect from the invasion of foreign organisms [104], foes or cheaters, which may trigger dysbiosis and collapsing of a well-ordered community. From a holobiont perspective, it is also important to consider that the equilibrium is guided by the host tolerance which can be reciprocated by the beneficial role of microorganisms such as in the digestion, brain-gut axis or plant growth promotion [105]. There are numerous combinations of strategies for competition between microorganisms, from nutrient scavenging to killing, from a distance or by contact [4]. Since most organisms will be using one or more of these tools, the competition outcome is dependent on how effective any one weapon is, and which combinations are the most potent.

Here we have conducted a comprehensive and systematic evaluation of potency/role of every single *P. aeruginosa* T6SS toxins. Table 1 lists all currently known toxins, and there is a clear incentive towards the idea that this is yet the tip of the iceberg with many more to be found [38,53]. It is puzzling though why so many toxins would be needed to eliminate a single prey. One may think that avoiding emergence of T6SS resistance would be one reason, but others might be synergy and differential effectiveness [38]. Here, that is thus *ca*. 20 strains that have been engineered so that each strain is theoretically susceptible to a single T6SS toxin. These toxins are mainly delivered by the H1- and H2-T6SS since in case of the H3-T6SS, the best characterized substrates are involved in ion uptake, *e.g.* TseF [89], but not in direct antagonism. Our data demonstrate that every single toxin counts, and, in most cases, the sensitized prey is limited in its expansion by the attacker within a macrocolony context. In the cases where the impact is not obvious from the macroscopic analysis an adequate toxin combination or a change in conditions could unveil their contribution.

A striking example of synergy provided in our study is with the two peptidoglycan hydrolases released by the H1-T6SS, Tse1 and Tse3. Individually sensitized strains in a background deleted for the *rsmA* gene are not challenged for their expansion when in contact with attacker cells. However, in this same genetic background, the dual sensitization against Tse1 and Tse3 simultaneously, is now showing a very clear prey growth restriction pattern (Fig 4). It suggests that the disruption of both the glycan backbone and the peptidyl bonds in between the glycan chains are needed to collapse the sacculus structure. However, the synergy might be needed only if each enzyme is delivered at very low dose. Such hypothesis could be verified when another genetic background is used, for which we showed that the H1-T6SS production and firing dynamic is higher as compared to the Δ*rsmA* context, *e.g.* Δ*retS*. In this case the strain sensitized to Tse3 is readily challenged by the attacker, although the Tse1-sensitized strain is still resistant (Fig 4B). This is an important observation showing that the dose of toxin injected, and thus the firing efficacy is an important parameter in the outcome of competition. High firing rate allows to overcome limitations posed by low toxicity impact of a given toxin, and from this we might conclude that Tse3 is more potent than Tse1. The firing efficacy also helps in resolving the contribution of many toxins delivered by the H2-T6SS and notably the multiple lipases and phospholipases. Whereas Tle3 and PldB sensitization is readily visible using a Δ*rsmA* background, for other phospholipases, notably PldA and Tle1, slight impact is only

visible in a Δ*retS* background (S12 Fig). Furthermore, and in this case that is valid for all H2-T6SS toxins, the sensitivity is clearly observed at 25˚C and not at 37˚C (S10 Fig), which contrast with H1-T6SS toxins. It was previously reported that particular environmental conditions, *e.g.* osmolarity, pH or temperature, may potentiate the activity of a subset of toxins [38], here it is more likely that the injection dose is dependent on the temperature as observed by the level of expression of these two systems at these different temperatures.

We also used combination of experimental and modelling approaches to confirm that high frequency of toxin injection depends not only on the activity of the system but also on the number of contacts between attackers and preys. The effects of inoculum content on population intermixing have been shown in previous studies both for CDI and T6SS [39,40,92,93]. These studies mainly establish the role of lytic toxins and advantages between constitutive and retaliation T6SS firing strategies but from a mechanistic point of view T6SS fights might involve a multitude of additional parameters. The contact indeed does not only involve where two cells touch each other, but also whether a T6SS is assembled at this point of contact. We have previously observed that in a Δ*rsmA* background there are many more H2-T6SS in a cell as compared to H1-T6SS [35]. We have also proposed that the H1-T6SS is far more rapid at assembling and injecting as compared to the H2-T6SS [51]. Finally, it was shown that the H1-T6SS operates as a retaliation system, which means responding only upon T6SS attack from an opponent and at the point of contact [106], which is unlikely the case for H2-T6SS. What would be a better competitive strategy is again a matter of conditions and context. A recent study has proposed that the retaliation, which might intuitively seem not effective in the face of a strong opponent, can be beneficial on the long term [107]. The model presented in the study suggests that it is in fact very much dependent on how many times the bacterium can fire back once triggered. If that is many times and quick, then it would be cost effective and beneficial. The systematic use of the T6SS might be energetically costly if fired in circumstances where it is not desperately needed. Some bacterial species have evolved compromises here, in which they kept arrays of immunity genes [108,109], while no longer using the T6SS or its toxin, so being defensively equipped to be part of an offensive community.

Since the contact is instrumental for T6SS-dependent competition, determinants that are important in promoting contact events might thus influence drastically the outcome. T4P are proteinaceous fibres extending from the bacterial surface, that are used for both cell-cell adhesion and cell-surface adhesion [94]. Both of these functions are mediated through the same mechanism–dynamic pili polymerisation and de-polymerisation that allows bacteria to exert force against the object that it adheres to [110]. In case of T4P-mediated cell-cell interactions, adjacent bacteria form aggregates through pili-pili interactions. When T4P adhere to a substrate, this same mechanism is used to propel bacteria across surface through a type of motility termed "twitching" [111]. Here we showed that lack of T4P-dependent mediated contact between cells allows prey to better resist T6SS attack, likely because of loosen contacts. Strikingly, lack of T4P in the attacker, while the prey is motile, promotes prey escape which results in mixed colonies with a nice crown of prey "escapers" around a uniform centre of T6SS attacker. This is particularly original, since a similar study on impact of T4P on sub-population distribution in whole mixed colonies focusing on *Neisseria gonorrhoeae* shows the opposite effect. Indeed, in a mix of T4P+ and T4P- strains, *N. gonorrhoeae* lacking pili are found in the colony outer region [112]. This has been attributed to a somewhat ubiquitous adhesion-based sorting mechanism resulting in phase separation [113]. Specific cell-cell adhesion-mediated sorting, where non-adherent sub-population is expelled to the outer population boundary appears to be a biophysical mechanism observed in normal organogenesis, tumorigenesis [114] and organisation of bacterial populations [112,115]. Furthermore, another study assessing T4P impact on *Neisseria cinerea* T6SS-mediated killing observed that sub-population of

cells that loose T4P escapes otherwise proficient T6SS killing. In this species, it is likely that the presence of T4P does not allow segregation and contact avoidance from the T6SS attacker [116]. Finally, a previous study looking at T4P contribution to *P. aeruginosa* swarming has also noted that in a mix of T4P+ and T4P- bacteria, the strains lacking pili are preferentially found at the leading edge of a swarming population [117]. Altogether it thus appears that presence/absence of T4P, specific growth conditions and T6SS-dependent effective competition, play a role in determining sub-population distribution within a bacterial community and this may drastically vary from one bacterial species to another.

Every study model, both experimental and theoretical, is limited by the fact that only specific environmental conditions and a restricted number of parameters are considered. As such making predictions on the outcome of a competition happening in a host like the lungs of cystic fibrosis patients remains a real challenge [118]. *P. aeruginosa* in young CF patients is not necessarily the main resident in the lungs but becomes dominant in adults thus the interest in addressing which mechanisms is used to gain advantage in this niche. The full complexity of natural communities is unlikely to be recapitulated in minimal systems such as the one we used here, and competition outcome likely does not depend on one strategy but on the sum of many. Yet we could evaluate the impact of individual T6SS toxins or T4P-dependent contact during competition. We believe that recording the contribution of individual elements, and that is one major advance provided by studies as the one we conducted here, is essential for construction of large and comprehensive datasets. These can be used in conjunction with theoretical tools to gain further understanding on what rules govern outcome of competition. If the T6SS is a key machine in the resolution of microbial competition, genomic approaches showed that despite a broad distribution across phyla and species, there is no rule on who does or does not pose a T6SS [119]. Trying to figure out the benefit for bacteria of having or not a T6SS would only make sense if we understand which toxins are delivered by these systems, and what are their impact on preys. Further studies should also assess the distribution of cell lineages within the three dimensions of a biofilm [120] which will further complexify the heterogeneity of the environment and how cells respond and adapt within the different layers of a biofilm.

## Materials and methods

### Bacterial strains and growth conditions

*P. aeruginosa* and *E. coli* strains used in the study are listed in S1 Table. Plasmids used in the study are listed in S2 Table. Unless otherwise specified bacteria were cultured in lysogeny broth (LB) (Miller) with agitation or on LB agar (Miller) plates at 37˚C. When appropriate following antibiotics were supplemented at given concentrations–streptomycin (50μg/mL), kanamycin (50μg/mL), gentamicin (25μg/mL), tetracycline (15μg/mL) for *E. coli* and streptomycin (2000μg/mL), gentamicin (50μg/mL), tetracycline (150μg/mL) for *P. aeruginosa*.

### DNA manipulation

For purposes of use as template for construct cloning, *P. aeruginosa* PAO1 genomic DNA was used after isolation using PureLink Genomic DNA minikit (Invitrogen). Plasmids were purified using QIAprep spin miniprep kit (Qiagen). Oligonucleotides used for gene cloning or screening purposes were purchased from Sigma and their sequences are listed in the S3 Table. DNA fragments used for construction of mutators, and expression reporter plasmids were cloned using KOD Hot Start DNA Polymerase (Novagen). PCR and digestion products were purified using QIAquick PCR & Gel Cleanup kits (Qiagen). Zero Blunt TOPO kit (Invitrogen) was used for ligation and further amplification of PCR products. Restriction enzymes–*Apa*I,

*Bam*HI, *Blp*I, *Kpn*I, *Nsi*I, *Sal*I, *Sma*I, *Spe*I, *Sph*I by New England Biolabs were used, with all reactions performed according to manufacturer's instructions. Constructs were ligated using T4 DNA ligase (NEB) according to manufacturer's protocol. Screening of mutant strains was performed via colony PCR using Taq polymerase. Sequences of plasmid constructs were verified using sequencing performed by GATC Biotech.

### *P. aeruginosa* genomic manipulation and fluorescent tagging

Gene deletions and insertion of chimeric proteins were performed though allelic exchange using a suicide vector pKNG101 [121] resulting in targeted and scarless removal of genes of interest or addition of fluorescent fusion. Vector integration into genome is guided by approx. 500bp upstream and 500bp downstream regions flanking sequence of interest. Deletions were constructed in frame leaving 9bp from the beginning of the gene and 9bp from the end of the gene including the STOP codon (These fragments were extended to preserve function of surrounding genes in cases of overlapping open reading frames). The fragments mentioned were amplified using *P. aeruginosa* PAO1 genomic DNA as template and constructs assembled using splicing by overlap PCR.

Mutator plasmid were delivered to *P. aeruginosa* via three partner conjugation and incorporation in genome was selected for by growth on streptomycin containing VBM (Vogel-Bonner media) agar. Subsequently colonies are streaked on 20% (w/v) sucrose containing LB agar and grown at room temperature for 48 to 72h. As pKNG101 encodes *Bacillus subtilis* derived *sacB* gene that introduces sucrose sensitivity. This counter-selection process promotes excision of the vector, the resulting strains revert to the wild-type genotype or contain only the alternative allele without leaving any marks indicative of the process in the genome. Bacteria growing on sucrose containing media are replica-patched on LB and LBSm2000 and screened via colony PCR to confirm the strain genotypes.

*P. aeruginosa* fluorescent tags are chromosomally integrated and constitutively expressed under pX2 promoter. miniCTX plasmids containing sfGFP (pNUT1295) and mCherry (pNUT1108) were delivered into *P. aeruginosa* using three partner conjugation.

### Measuring gene expression using plasmid-based reporters

Plasmid based transcriptional and translational reporter fusions were used to asses gene expression levels. Promoter fragment region of +30bp to -500bp upstream of *tssA1*, *tssA2* and *tssB3* were amplified by PCR using *P. aeruginosa* PAO1 genomic DNA as template. Promoter fragments were placed into a pUCP22 plasmid [122] upstream of a gene encoding a stable green fluorescent protein variant (GFPmut3b) [47]. Translational reporters were designed such that 10 leading amino acids encoded from the *tss* gene were translationally fused to *gfp*, such that fluorophore translation occurs from the same RBS as the *tss* genes. For transcriptional reporters, the *tss* promoter is used and an additional RBS has been included to allow for GFP production independently of translation of the given *tss* gene. All plasmids were validated by sequencing. For purposes of measurements of promoter activity, pUCP22 based reporter plasmids were delivered to *P. aeruginosa* strains using electroporation, and plasmid containing stains were selected for by growth on LB Gm50.

Fluorescent reporter measurements were performed by growing *P. aeruginosa* strains in 48-well plates without agitation. *P. aeruginosa* overnight cultures were harvested by centrifugation (3min 8000rpm) and both supernatant and extracellular matrix (ECM) layer were discarded. Biofilm-free pallet was washed twice by re-suspending in Phosphate-buffered saline (PBS) and subsequent centrifugation. After washing, pallet was resuspended in fresh PBS and bacterial concentration adjusted to $OD_{600}$ 1.0. 48-well plates containing 360μL of LB+ Gent50

were inoculated with 40μL of bacterial culture. Promoter activity measurements were performed while growing culture in FLUOstar OMEGA plate reader at 37°C and 25°C. OD measurements were performed at 600nm and GFP measurements used 485nm excitation and 520nm emission filter.

## T6SS expression and secretion sample preparation

For the purposes of analysing T6SS component expression and secretion strains of interest were grown overnight and sub-cultured in tryptic soy broth (TSB) at staring concentration of $OD_{600}$ 0.1. For analysis of H1-T6SS strains were cultured at 37°C for 6h, for H2-T6SS at 25°C for 18h and for H3-T6SSS at 30°C for 24h. Cell lysate was prepared by harvesting culture volume equivalent to 1mL of $OD_{600}$ = 1.5, pelleted by centrifugation (3min, 8000rpm) and resuspended in 150μL of Laemmli buffer. Preparation of secretion samples was performed as described previously [123]. Briefly, cell free supernatant was acquired by repeated centrifugation (20min, 4000g, 4°C). Proteins from cell free supernatant were precipitated using 10% trichloracetic acid. Protein pellets were washed with 90% acetone, air dried and resuspended in Laemmli buffer. Both cell lysate and supernatant samples were boiled at 95°C for 10min before loading onto SDS-PAGE gels for analysis via western blotting.

## Western blot analysis

For data shown in Figs 1 and S2, samples were separated using 12% acrylamide [0.375M Tris (pH 8.8), 0.1% SDS, 0.1% ammonium persulfate, 0.4% TMED] with a stacking gel of 5% acrylamide [0.265M Tris(pH6.8), 0.1% SDS, 0.1% ammonium persulfate, 0.1% TMED] and run in Tris-glycine-SDS buffer [0.3%(w/v) Tris, 1.44%(w/v) glycine, 0.01% SDS, pH 8.3] at 80V for 20min and at 130V until desired band separation achieved. Precision Plus Protein Kaleidoscope Prestained Protein Standard (BioRad) was used as molecular weight standard. For western blots, transfer to nitrocellulose membranes (Amersham Protran Western blotting membranes, pore size 0.2 μm) was performed at 24V, 0.18A for 45min using transfer buffer [10% Tris-glycine-SDS buffer, 20% ethanol]. Membrane blocking was performed at 4°C overnight in blocking buffer [5% (w/v) milk powder in 1xTBST (50mM Tris, 150mM NaCl, pH 8.0, 0.1% Tween20)]. Polyclonal antibodies against Hcp1, Hcp2, and Hcp3 were used at 1:500 working concentration. Monoclonal antibodies were used against RNA polymerase β subunit at 1:5000 dilution (RpoB, NeoClone). Membranes were washed with TBST before addition of HRP-conjugated secondary antibodies (Sigma), anti-mouse was used for RpoB and anti-rabbit for the Hcp1, Hcp2 and Hcp3. Blots were developed using Novex ECL Chemiluminescent Substrate reagent mix (Millipore) for approx. 1min and membranes were imaged using BioRad ChimiDoc.

For data shown in S18 Fig, *P. aeruginosa* strains were grown overnight in tryptic soy broth (TSB) at 37°C. Bacterial cultures were then subcultured to an $OD_{600}$ of 0.1. For H1 T6SS cultures were grown for 5–6 h at 37°C, for H2 T6SS cultures were grown for 8 h at 25°C and for H3 T6SS cultures were grown for 24 h at 25°C. After adjusting the bacterial culture to $OD_{600}$ of 1, 1 ml was harvested and resuspended in 100 μl 1.7xLaemmli buffer to prepare the whole cell lysate. 13 ml of same culture was centrifuged at 4000g for 20 min at 4°C to separate bacterial cells from supernatant. Then 10 ml, and subsequently 7ml, of supernatant was transferred to another falcon tube and centrifuged again. Finally, 200 μl of 100% TCA was added to 1.8 ml of the supernatant and proteins were precipitated overnight at 4°C. The protein precipitate was centrifuged at 16000g for 30 min at 4°C. The pellet was then washed with ice cold 90% acetone and centrifuged, then air dried for 30 min and resuspended in 1.7xLaemmli buffer to an OD equivalent of 20 to prepare the secreted protein sample. Both cell lysate and secreted

samples were boiled at 100˚C for 10 min before resolving on SDS-PAGE gels. The samples were separated using a 15% resolving gel and 4% stacking gel in Tris-glycine-SDS buffer. Gels were run at 80V at constant current. Spectra Multicolor Broad Range protein ladder (Thermo-Fisher Scientific, 26634) was used as the molecular weight standard. Separated protein samples were then transferred to PVDF membranes using iBLOT2 dry transfer apparatus (Thermo-Fisher Scientific) at 20V for 7 min. The blots were then stained with Poinceau S to visualize protein transfer. The stain was removed by washing with TBST (50 mM Tris, 150 mM NaCl, pH7.5, 0.1% Tween-20). The blots were blocked in 5% skimmed milk in TBST for 1 h. Poly-clonal antibodies against Hcp1, Hcp2 or Hcp3 were then incubated with the blots at 1:500 dilution in blocking solution for 1 h. The blots were washed twice with TBST for 5 min each and incubated with anti-rabbit HRP-conjugated secondary antibody (Sigma-Alrich, A0545) for 1 h. The blots were then washed twice with TBST and incubated with Super Signal West Pico PLUS Chemiluminescent substrate (ThermoFisher Scientific, 34577) for 5 min and imaged using Amersham ImageQuant800 (GE).

## Twitching assay

Twitching motility was assayed using freshly poured 10mL (in standard petri dish) LB plates with 1% (w/v) agar. Plates were inoculated by using a pipette tip to pick a colony from a freshly grown LB agar plate and stabbing the centre of the twitching plate. Plates were incubated at 30˚C for 24h. After removing agar bacteria attached to the plate stained with 1% (w/v) crystal violet solution. After staining for 3-5min, plates were gently washed using distilled deionised water, twitching zones were subsequently imaged and measured.

## Single cell and whole colony microscopy

To prepare samples for microscopy at single cell resolution bacterial overnight cultures were pelleted (3min, 8000rpm). After discarding supernatant and ECM fraction bacteria are washed by resuspending in PBS and subsequent centrifugation. Clean bacterial pallets were resuspended in PBS and concentration adjusted to $OD_{600}$ = 0.05. 7μL of sample are spotted on LB agar pads and dried. LB agar with dried bacteria is inverted onto μ-Dish 35 mm glass-bottom microscopy dishes (Ibidi). Before imaging samples are incubated at 37˚C for 4h.

For purposes of whole colony imaging, *P. aeruginosa* strains with constitutively expressed chromosomal mCherry or sfGFP tags were used. To prepare bacterial mixes, overnight cultures were pelleted (3min 8000rpm), supernatant and ECM fraction were discarded. Bacterial pallets were washed by resuspending in PBS and subsequent centrifugation (8000rpm 3min) twice. After washing bacterial pellets were resuspended in fresh PBS and cell density is adjusted to $OD_{600}$ 1.0. Prey and attacker bacteria are mixed at 1 to 1 ratio (unless investigating effects of mixing ratio on the competition, where bacteria were mixed at ratios specified in the figure legend). Bacterial mixes were diluted depending on the specific setup and 1μL of each mixed culture were spotted on LB agar (1.2% w/v agar when growing bacteria at 25˚C and 2.0% w/v when culturing at 37˚C). Inoculum spots were dried and incubated for 48h at 37˚C for H1-T6SS dependent competition and at 25˚C for H2-T6SS dependent competition (unless otherwise specified).

Images were acquired using Zeiss Axio Observer 7 microscope, under Zeiss Colibri 7 illumination, using Hamamatsu Flash4 camera and environmental control was used to maintain incubation temperature during imaging. Plan-Apochromat 100x 1.40 Oil Ph3 M27 objective was used for single cell imaging. For imaging whole bacterial colonies EC Plan-Neofluar 10x/0.30 Ph1 or EC Plan-Neofluar 5x/0.16 Ph1 M27 objectives were used. All imaging was

performed at the Facility for Imaging by Light Microscopy (FILM) at Imperial College London, UK.

## Complementation of T6SS toxins sensitized strains

Some of the mutant strains sensitized against a given T6SS toxin were complemented by reintroducing the cognate immunity gene in trans. The full length *tsi2* (234 bp), *tsi5* (231 bp) and *tsiT* (720 bp) genes, along with Shine-Dalgarno sequences were amplified from *P. aeruginosa* wild type using primers listed in S3 Table. The PCR temperature cycling conditions were as follows: initial denaturation at 98°C for 30 s followed by 35 standard cycles of denaturation at 98°CC for 10 s, primer annealing at 69°C for 30 s and primer extension at 72°C for 30 s. All PCR fragments were cloned into pBBRMCS-5 vector at *Eco*RI and *Xba*I restriction sites to generate pBBR*tsi2*, pBBR*tsi5* and pBBR*tsiT*. The presence of wild type copies of genes was confirmed by sequencing (Biobasic, Singapore). The pBBR*tli3*-HA construct was available from Filloux lab strain collection. The mCherry-tagged strains carrying mutations in the effector-immunity genes in Δ*retS* (for *tsi2* and *tsi5*) or Δ*rsmA* (for *tli3* and *tsiT*) background were transformed with either empty pBBRMCS-5 vector or immunity gene constructs (pBBR*tsi2*, pBBR*tsi5*, pBBR*tli3* or pBBR*tsiT*) by electroporation (25 μF, 200 Ω and 2.5 kV cm$^{-1}$) using a Gene Pulsar (Biorad, Hercules, CA, USA). The attacker strains, either Δ*retS* or Δ*rsmA* tagged with sfGFP, were also transformed with empty pBBRMCS-5 vector. The transformants were selected on gentamicin containing medium. Subsequently the immunity gene complementation impact was assessed by whole colony imaging. Overnight bacterial cultures of attacker and prey strains grown in the presence of gentamicin were pelleted (8000 rpm, 3 min) and supernatant was discarded. The bacterial pellets were resuspended in PBS and washed twice similarly. The cell suspensions in PBS were adjusted to an $OD_{600}$ of 1.0. Equal volumes of prey and attacker cells were mixed and 1 μl of each mixed culture was spotted on 0.5 ml LB agar containing gentamicin in 24-well plates (1.2% w/v agar when growing bacteria at 25°C for H2 T6SS dependent competition and 2.0% w/v when culturing at 37°C for H1 T6SS dependent competition). Plates were dried and incubated at appropriate temperatures for 48 h. Tile scanning was carried out to image the whole colony using Carl Zeiss Axio Observer Z1 inverted microscope with an EC Plan-Neofluar 5x/0.16 dry objective (Carl Zeiss, Singapore). The images were captured and processed using Zeiss Zen 2.3 software (Carl Zeiss).

## Biophysical simulations of bacterial colonies

Agent-based modelling framework—CellModeller [124]—was used with T6SS interactions subsequently implemented as described previously [125]. Briefly, CellModeller is an agent-based biophysical modelling framework used to simulate 2D representation of microcolonies by explicitly describing bacterial growth at a single cell level based on growth, volume exclusion, and adjustable cell-cell interactions. Selected key details are highlighted below and key simulation parameters listed in S4 Table.

## Cell growth and movement

Within the simulation framework, bacterial communities are represented as collections of rod-shaped rigid cells, whose growth occurs via elongation until a set target size is reached. Subsequently division via binary fission occurs producing 2 identical daughter cells. It is assumed that a constant source of nutrients is available, therefore bacterial growth is limited only by lack of physical space or via direct negative interactions. Bacteria simulated do not aggregate and do not move actively, but cell arrangement changes due to physical forces

exerted between individual cells during growth and because of random noise perturbations after division events to ensure no overlap between cells.

### T6SS interactions

T6SS firing is described as spatially explicit events. Each of the T6SS active cells can fire a set number of needles that are randomly distributed across the attacker cell. Number of firing events for each individual bacteria in each timestep are drawn from Poisson distribution where mean number of needles is parameterized accordingly to the specific setup. Each firing event carries an associated metabolic cost, specific maximum cell growth rate is reduced to "pay" set material penalty for each firing event. For purposes of simulations here, T6SS firing costs were set as negligibly low, as this has been shown to affect the specific optimum firing rate, but not the general "rules of engagement" and broader effect trends that were of interest here [125].

 T6SS hit detection has been previously described in detail [107]. Briefly, a test is performed to check whether a needle geometrically intersects any cell except for the one firing it. All T6SS hits within a simulation setup are assumed to deliver constant amount of identical toxin. Toxin mode of action is described by toxicity (Toxin lethal dose), that determines the total dose of toxin reacquired to kill a sensitive cell, partial intoxication leads to proportional reduction in maximum prey growth rate. Another toxin characteristic is "lysis delay" that determines the time between cell death and the cell being removed from the simulation. As these setups allow for partial intoxication, gradual prey recovery is implemented, where prey cell's internal toxin pool undergoes exponential decay over time, but the type of recovery implemented has been shown to have little effect on contact-based interactions, as non-motile cells are unlikely to lose contact [39].

### Model setup

Initial simulation steps are set-up to reflect a scaled down inoculation of mixed bacterial colony experiments. A mix of prey and attacker are randomly placed within a 110μm radius circular area at a ratio and initial density determined by the specific setup of interest. Growth and interactions in these simulations were restricted to 2D plane and run for a set number of timesteps (equivalent of 25h). The simulation run execution time of examples shown in results section ranges between 15min and 40min (when testing on laptop with AMD Ryzen 9 5900HS and NVIDIA GeForce RTX 3060) and is largely dependent on number of individual bacteria described and number of T6SS needles being fired. 10 simulation runs were performed for each set of parameters.

### Calculation of contact assortment

Contact assortment indicates how intermixed or segregated are sub-populations within simulation and was adapted from previous work [93]. Contact assortment measures how intermixed are different types of cells within community as a whole considering direct cell-cell contacts in simulation, rather than assortment of species in a local area with a set radius. Mean contact assortment trends towards 1 when highest number of inter-population contacts are present and towards 0 when populations spatially segregate and few interspecies contacts are present. For each of the cells average number of interspecies contacts per total number of contacts was calculated. Subsequently, mean interspecies contact frequency for prey cell was divided by global prey to attacker ratio.

## Image analysis

Calculation of intermixing index was adapted from previous studies [126,127]. Briefly, intermixing index is defined as number of intersection events between the 2 populations in a circular section at a given distance from colony centre, correcting for circumference with a given radius. After image tile stitching performed using ZenBlue (Zeiss), colony central points are determined manually using Fiji (ImageJ). Subsequently, histogram equalized images are used to calculate relative signal intensity at a given radius from colony central point for each of the fluorescence channels, this was used to produce a binarized matrix describing which of the strains is more abundant at a given location. (Relative signal intensity was normalised to colony central region.) This binarized matrix was used to quantify number of intersection events between the 2 strains at a given distance from colony central point. Number of intersection events was normalised for circumference at a given radius to produce intermixing index. Elevation in intermixing index towards 1 indicates that both populations are equally abundant and distribution alternates, while trend towards 0 indicates that only one of the species is present or species are comparably more spatially segregated.

After performing histogram equalization and manually determining the central points of the images, we determined the mean fluorescence signal intensity and its variation at different distances from the central point. To do this, we selected a set of pixels at a given distance from the central point and used these values to calculate the mean signal intensity and variance as a function of distance from the colony center. We then compared the normalized channel brightness of each species to determine the more abundant one. For spatially explicit relative prey abundance analysis, we performed this comparison for sets of pixels at different distances from the colony center. To calculate the overall relative prey abundance, we performed the analysis for the entire outer colony region. We then used these values to test for statistical differences in relative prey abundance between T6SS+ and T6SS- parental strains. Custom image analysis and statistical testing were performed using Python 3.

## Data processing

Initial image processing including shading correction and tiling as well as histogram adjustment for purposes of display is done using ZEN lite (Zeiss). Single cell data analysis, mixed colony dimension measurements and further manual image analysis done using Fiji (ImageJ). Analysis of spatial fluorescence signal distribution in mixed bacterial colonies and calculation of intermixing and assortment indices as well as modelling result analysis, summary statistics and all experimental data plotting done by using Python3.

## Supporting information

**S1 Fig. Mutant strains in the Gac/Rsm pathway show graduated increase in H1-T6SS and H2-T6SS promoter translational activity.** Analysis of H1-T6SS (**A**) and H2-T6SS (**B**) promoter translational activity in WT, Δ*rsmN*, Δ*rsmA*, Δ*retS*, and Δ*rsmA*Δ*rsmN* strains. Measurements performed in static biofilm using plasmid based GFPmut3b reporter fusions. All measurements performed at 37˚C, in static culture, each display item shows a mean +SD of 4 technical replicates that is representative of n = 3 biologically independent repeats. (TIF)

**S2 Fig. Mutant strains in the Gac/Rsm pathway show graduated increase in H3-T6SS promoter activity and expression.** Analysis of H3-T6SS promoter transcriptional (**A**) and translational (**B**) activity in WT, Δ*rsmN*, Δ*rsmA*, Δ*retS*, and Δ*rsmA*Δ*rsmN* strains. Measurements performed in static biofilm using plasmid based GFPmut3b reporter fusions. All

measurements performed at 37˚C, in static culture, each display item shows a mean +SD of 4 technical replicates that is representative of n = 3 biologically independent repeats. (**C**) Western blot analysis shows gradual elevation in of Hcp3 expression in WT, Δ*rsmN*, Δ*rsmA*, Δ*retS*, and Δ*rsmA*Δ*rsmN* strains. A representative blot of 3 independent biological repeats shown here, for H3-T6SS activity assessment bacteria were cultured for 24h at 30˚C. RNA polymerase (RpoB) used as loading control.
(TIF)

**S3 Fig. Decrease in growth temperature results in elevated H2-T6SS and H3-T6SS promoter activity, while H1-T6SS promoter activity is decreased in mutant strains altered in the Gac/Rsm regulatory cascade.** Analysis of T6SS promoter transcriptional (left) and translational (right) activity in WT, Δ*rsmN*, Δ*rsmA*, Δ*retS*, and Δ*rsmA*Δ*rsmN* strains. Upper row shows H1-T6SS (*tssA1*), middle H2-T6SS (*tssA2*), and lower H3-T6SS (*tssB3*), promoter activity over growth time as measured by plasmid based GFPmut3b reporter fusions. All measurements performed at 25˚C, in static culture, each display item shows a mean +SD of 4 technical replicates that is representative of n = 3 biologically independent repeats.
(TIF)

**S4 Fig. Changes in distribution of individual sub-populations in macrocolonies of muatnts altered in the Gac/Rsm pathway depending on inoculum density.** Single channel images corresponding to composite in Fig 2A–mCherry in red (**A**) and sfGFP in green (**B**). Mixed bacterial colonies of WT, Δ*rsmA*, Δ*retS*, and Δ*rsmA*Δ*rsmN* strains with altered inoculum densities. Isogenic bacterial strains tagged with mCherry (red) and sfGFP (green) fluorophores were mixed at 1 to 1 ratio and after adjusting inoculum density (OD$_{600}$ = 1.0; 0.1; 0.01; 0.001) spotted on LB agar, images of whole microcolonies taken after 48h incubation at 37˚C show 2 morphologically distinct regions—highly mixed inner region corresponding to inoculum zone and outer region where spatial segregation of the sub-populations is apparent.
(TIF)

**S5 Fig. Increase in fluorescence signal variance and decreased intermixing are indicative of sub-population segregation in outer colony regions.** Analysis of spatial fluorescent signal distribution for fluorescence images shown in Fig 2. (**A**) Relative mean signal intensity + SD of individual fluorescence channels at different distances from the colony centre. (**B**) Relative sub-population intermixing at different distances from the macrocolony centre, calculated for circular sections taken at increasing distance from colony central point.
(TIF)

**S6 Fig. Loss of the ability to fire T6SS does not result in loss of competitive fitness in isogenic strains.** Fluorescent images of whole bacterial colonies of bacteria mixes made up of T6SS+ (Δ*retS*) and T6SS–(Δ*retS*Δ*tssB1*Δ*tssB2*Δ*tssB3*) strains at 1:1 initial ratio and initial OD$_{600}$ = 1.0, showing individual channel and overlay images of 3 biological repeats.
(TIF)

**S7 Fig. Further elevation in H1-T6SS activity enables further toxin sensitised prey elimination.** Representative image of 48h old mixed colonies of toxin sensitised bacteria in red in competition with T6SS+ or T6SS- (Δ*tssB1*Δ*tssB2*Δ*tssB3*) strains of the same regulatory background in green. Upper lane contains a control mix of bacteria with full toxin-immunity gene sets, each of the following lanes contains strain sensitised to one of the H1-T6SS toxins from Tse1 to Tse8. Images sets of competitions of WT(**A**) and Δ*rsmA*Δ*rsmN* (**B**) background strains shown with each of the sets containing both single fluorescence channel and overlay images showing distribution of sensitised prey in a mix with T6SS$^{+}$ and subsequently T6SS-

(Δ*tssB1*Δ*tssB2*Δ*tssB3*) parental strain. Strains contain constitutively expressed fluorescent proteins, prey labelled with mCherry (shown in red) and attacker with sfGFP (shown in green). All bacteria mixed at 1:1 ratio, inoculum $OD_{600}$ = 1.0, grown for 48h at 37˚C on LB with 2% (w/v) agar.
(TIF)

**S8 Fig. T6SS toxin action impacts bacterial distribution within communities.** (**A**) Representative images of mixed bacterial macrocolonies consisting of H1-T6SS toxin sensitised bacteria (in red) in presence of T6SS+ (Δ*rsmA*) or T6SS- (Δ*rsmA*Δ*tssB1*Δ*tssB2*Δ*tssB3*) parental bacteria (in green). Upper lane contains a control mix of bacteria with full toxin-immunity gene sets, each of the following lanes contains strain sensitised to one of the H1-T6SS toxins from Tse1 to Tse8. Fluorescent channel overlay images show bacterial distribution within 3 replicate colonies. All bacteria mixed at 1 to 1 ratio with inoculum density of $OD_{600}$ = 0.01, grown for 48h at 37˚C on LB with 2% (w/v) agar. (**B**) Images showing distribution of individual strains corresponding to replicates #1 from panel (**A**) and the corresponding analysis of fluorescent signal distribution within the image. Columns 2 and 5 show the corresponding mean (+SD) relative fluorescence signal intensity for the each of the channels as function of distance from the colony central point (for image replicates #1). With columns 3 and 6 showing relative area occupied by prey as function of distance from the colony central point (mean of 3 replicate colonies +SD).
(TIF)

**S9 Fig. Statistical analysis of relative decrease in prey abundance in presence of T6SS + parental strain.** Summary analysis of images from S8 Fig, Measurement of relative outer colony region occupied by toxin sensitised prey in presence of T6SS+ (Δ*rsmA*) or T6SS- (Δ*rsmA*Δ*tssB1*Δ*tssB2*Δ*tssB3*) parental bacteria. Mean (+SD) of 3 biological replicates, statistical significance measured using students T-test, where $p < 0.05$ indicated with a "\*". P-values for each of the prey strains as follows– Δ*rsmA*– 0.6238; Δ*rsmA*Δ*tse1tsi1*–0.0885; Δ*rsmA*Δ*tse2tsi2*– 0.0163; Δ*rsmA*Δ*ts31tsi3*–0.1450; Δ*rsmA*Δ*tse4tsi4*–0.0009; Δ*rsmA*Δ*tse5tsi5*–0.0474; Δ*rsmA*Δ*tse6tsi6*–0.0101; Δ*rsmA*Δ*tse7tsi7*–0.0429; Δ*rsmA*Δ*tse8tsi8*–0.7214.
(TIF)

**S10 Fig. H2-T6SS toxin mediated killing is observed when bacteria are grown at reduced temperature.** Sensitivity to H2-T6SS toxins can be observed when growing colonies at 25˚C, but not 37˚C. Representative images of 48h old mixed colonies of toxin sensitised Δ*rsmA* bacteria in red in competition with T6SS+ (Δ*rsmA*) or T6SS- (Δ*rsmA*Δ*tssB1*Δ*tssB2*Δ*tssB3*) parental strain in green. Upper lane contains a control mix of bacteria with full toxin-immunity gene sets, each of the following lanes contains strain sensitised to one of the H2-T6SS toxins in the following order: Tle1, Tle3, Tle4, PldA, PldB, TseT, TseV, VrgG2b, AmpDh3, PA5265, and common good effector Azu. Image sets of competitions show both single fluorescence channel and overlay images depicting distribution of sensitised prey in a mix with T6SS+ and subsequently T6SS- (Δ*tssB1*Δ*tssB2*Δ*tssB3*) parental strains at 2 different growth temperatures– 37˚C and 25˚C. Strains contain constitutively expressed fluorescent proteins, prey labelled with mCherry (shown in red) and attacker with sfGFP (shown in green). All bacteria mixed at 1:1 ratio, inoculum $OD_{600}$ = 1.0, grown for 48h, at 37˚C bacteria incubated on LB with 2% (w/v) agar, and at 25˚C bacteria incubated on LB with 1.2% (w/v) agar.
(TIF)

**S11 Fig. Expression of toxin immunities protects toxin sensitised bacteria from killing by T6SS proficient parental attacker.** Representative mixed colony images showing protective impact of toxin immunity complementation in both H1-T6SS (**A**) and H2-T6SS (**B**) toxin-

dependent competition. Showing individual fluorescence channel and overlay images of toxin sensitised prey (in red) carrying empty vector plasmid (left) and a plasmid expressing cognate immunity. (**A**) Tse2 ($\Delta retS\Delta tse2tsi2$) and Tse5 ($\Delta retS\Delta tse5tsi5$) sensitive prey bacteria in competition with T6SS+ ($\Delta retS$) parental strain. Bacteria mixed at 1 to 1 ratio, with inoculum density $OD_{600}$ = 1.0, grown for 48h at 37˚C on LB with 2% (w/v) agar. (B) Tle3 ($\Delta rsmA\Delta tle3tli3$) and TseT ($\Delta rsmA\Delta tseTtsiT$) sensitised prey bacteria in a mix with T6SS+ ($\Delta rsmA$) parental strain. Bacteria mixed at 1 to 1 ratio, with inoculum density $OD_{600}$ = 1.0, grown for 48h at 25˚C on LB with 1.2% (w/v) agar. Individual images of 3 biological replicates shown. (TIF)

**S12 Fig. Increased H2-T6SS toxin mediated prey growth restriction can be observed in $\Delta retS$ strains.** Representative images of 48h old mixed colonies of toxin sensitised $\Delta retS$ bacteria in red in competition with T6SS+ ($\Delta retS$) or T6SS- ($\Delta retS\Delta tssB1\Delta tssB2\Delta tssB3$) parental strain in green. Upper lane contains a control mix of bacteria with full toxin-immunity gene sets, each of the following lanes contains strain sensitised to one of the H2-T6SS toxins in the following order: Tle1, PldA, PldB, TseT and AmpDh3. Image sets of competitions show both single fluorescence channel and overlay images depicting distribution of sensitised prey in a mix with T6SS+ and subsequently T6SS- ($\Delta tssB1\Delta tssB2\Delta tssB3$) parental strains. Strains contain constitutively expressed fluorescent proteins, prey labelled with mCherry (shown in red) and attacker with sfGFP (shown in green). All bacteria mixed at 1:1 ratio, inoculum $OD_{600}$ = 1.0, grown for 48h at 25˚C on LB with 1.2% (w/v) agar. (TIF)

**S13 Fig. Inoculum composition determines prey/attacker contact assortment within macrocolony.** (**A**) Representative simulation outputs showing changes species distribution resulting from variation in initial density and mixing ratio of the populations. (**B**) Changes in localised agent intermixing as interspecies contact assortment resulting from variation in initial density and mixing ratio of cells in simulation setup. (No T6SS interactions, mean of n = 5). (TIF)

**S14 Fig. Time resolved changes in interspecies contacts are determined by both inoculum density and initial ratio of strains in the mix. (A, D)** Total number of cells over simulation time-course. **(B, E)** Total number of contact cells over simulation time-course. **(C, F)** Interspecies contact assortment over 30 simulation time-course. **A, B, and C** effect of variation in initial density. **(D, E, F)** effect of variation in species mixing ratio. (No T6SS interactions, mean +SD of n = 5). (TIF)

**S15 Fig. Set of simulation outputs (A) corresponding to Fig 5C with non-lytic toxin-based interactions. Set of corresponding green-fluorescent channel images (B) for the Fig 5D and a set of images of preys from Fig 5D in a mix with an T6SS- attacker strain (C). (A)** Representative simulation outputs showing how decrease in initial bacterial density promotes prey (red) survival in a mix an attacker population (green) with differing mean T6SS firing rate within a context of non-lytic toxins. (Toxin lethal dose = 5). **(B)** Single channel images corresponding to composite in Fig 5D showing distribution of attacker population in green. **(C)** Single channel and composite images of the prey set from Fig 5D. In a mix with T6SS- attacker ($\Delta tssB1\Delta tssB2\Delta tssB3$) of corresponding regulatory background. Mixed bacterial colonies of WT, $\Delta rsmA$, $\Delta retS$, and $\Delta rsmA\Delta rsmN$ strains with altered inoculum densities. Isogenic bacterial strains tagged with mCherry (red) and sfGFP (green) fluorophores were mixed at 1 to 1 ratio and after adjusting inoculum density (OD = 1.0; 0.1; 0.01; 0.001) spotted on LB agar,

images of whole microcolonies taken after 48h incubation at 37°C on LB with (2% w/v) agar.
(TIF)

**S16 Fig. Set of corresponding green-fluorescent channel images (A) for the Fig 5E and a set of images of preys from Fig 5E in a mix with an T6SS- attacker strain (B). (A)** Fluorescent single (sfGFP) channel images of mixed colonies from Fig 5F showing distribution of T6SS+ attacker strains. **(B)** Tse5 sensitive prey spatial distribution in presence of T6SS-*(ΔtssB1ΔtssB2ΔtssB3)* parental competitor strain. Individual and overlaid channel images shown. Each of the columns correspond to WT, *ΔrsmA*, *ΔretS*, and *ΔrsmAΔrsmN* regulatory background strains in the given order. Each of the rows contains a set of representative images of colonies set up with a different prey to attacker ratio in the inoculum. Prey to attacker ratios from the top row are as follows: 1:1, 2:1, 3:1, 5:1, 10:1. (Strains contain constitutively expressed fluorescent proteins, prey labelled with mCherry (shown in red) and attacker with sfGFP (shown in green). All bacteria mixed at ratios specified, inoculum $OD_{600}$ = 0.01, grown for 48h at 37°C on LB with 2% (w/v) agar.
(TIF)

**S17 Fig. Loss of T4P interferes with H1-T6SS mediated killing in the same manner as H2-T6SS.** Tse5 mediated competition was used to assess impact on H1-T6SS mediated killing in a mix of T4P+ and T4P- bacteria. The panel contains individual red channel images showing toxin sensitive prey distribution in presence of T6SS+ (left) or T6SS- (right) attacker strain in the upper two rows. Corresponding overlay images showing distribution of both toxin sensitive prey (in red—mCherry) and T6SS attacker population in green (sfGFP) shown in the lower two rows. T4P+ prey (*ΔrsmA Δtse5tsi5*) strains in the upper row, with T4P- prey (*ΔrsmA ΔpilA Δtse5tsi5*) in the lower row. With T4P+ attacker in the first column and T4P- attacker strains in the second. All bacteria mixed at 1 to 1 ratio, with inoculum density $OD_{600}$ = 1.0, grown for 48h at 37°C on LB with 2% (w/v) agar. 1 of 1 biological repeat shown for all images containing *ΔpilA* mutant strains.
(TIF)

**S18 Fig. Loss of T4P has no detectable impact on T6SS protein expression and system activity.** Western blot analysis shows that Hcp1(**A**), Hcp2 (**B**), and Hcp3 (**C**) protein levels remain consistent within cell lysate independently of the presence of T4P (*ΔpilA*). As assessed through secretion assay, no differences in H1-T6SS (**A**) and H2-T6SS (**B**) system activity are detectable upon loss of T4P. Bacteria were cultured for 5-6h at 37°C (H1-T6SS), for 8h at 25°C (H2-T6SS) and for 24h at 25°C (H3-T6SS).
(TIF)

**S1 Video. H1-T6SS dynamics in *rsmA* deletion strain.** Gradual sheath assembly and firing visualised using *tssB1*::mScarlet-I chromosomal fusion at native locus of *ΔrsmA* mutant bacteria growing on agar surface. The field of view of approx. 15.6 x 13 μm, showing 30 timelapse frames captured every 4s.
(AVI)

**S2 Video. H1-T6SS dynamics in *retS* deletion strain.** Gradual sheath assembly and firing visualised using *tssB1*::mScarlet-I chromosomal fusion at native locus of *ΔretS* mutant bacteria growing on agar surface. The field of view of approx. 15.6 x 13 μm, showing 30 timelapse frames captured every 4s.
(AVI)

**S3 Video. H1-T6SS dynamics in *rsmA/rsmN* double deletion strain.** Gradual sheath assembly and firing visualised using *tssB1*::mScarlet-I chromosomal fusion at native locus of *ΔrsmA*

and Δ*rsmN* mutant bacteria growing on agar surface. The field of view of approx. 15.6 x 13 μm, showing 30 timelapse frames captured every 4s.
(AVI)

**S1 Table. List of strains used in this study.**
(DOCX)

**S2 Table. List of plasmids used in this study.**
(DOCX)

**S3 Table. List of primers used in this study.**
(DOCX)

**S4 Table. Simulation parameters.**
(DOCX)

## Acknowledgments

The authors would like to thank Knut Drescher for helpful discussion and Nicholas Yuen for helping design Fig 6.

## Author Contributions

**Conceptualization:** Marta Rudzite, Alain Filloux.

**Data curation:** Marta Rudzite, Sujatha Subramoni, Robert G. Endres, Alain Filloux.

**Formal analysis:** Marta Rudzite, Robert G. Endres, Alain Filloux.

**Funding acquisition:** Robert G. Endres, Alain Filloux.

**Investigation:** Marta Rudzite, Sujatha Subramoni, Alain Filloux.

**Methodology:** Marta Rudzite, Sujatha Subramoni.

**Project administration:** Alain Filloux.

**Supervision:** Robert G. Endres, Alain Filloux.

**Writing – original draft:** Alain Filloux.

**Writing – review & editing:** Marta Rudzite, Robert G. Endres.

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
