## [Decision Letter · Decision Letter 0]

5 Feb 2023

Dear Prof. Filloux,

Thank you very much for submitting your manuscript "Effectiveness of Pseudomonas aeruginosa type VI secretion system relies on toxin potency and type IV pili-dependent interaction" for consideration at PLOS Pathogens and your patience during the extraordinarily long review process. As with all papers reviewed by the journal, your manuscript was reviewed by members of the editorial board and by several independent reviewers. In light of the reviews (below this email), we would like to invite the resubmission of a revised version that takes into account the reviewers' comments. We invite you to pay particular attention to the requests of reviewers for quantifications of the observations. 

We cannot make any decision about publication until we have seen the revised manuscript and your response to the reviewers' comments. Your revised manuscript is also likely to be sent to reviewers for further evaluation.

Sincerely,

Sophie Helaine

Academic Editor

PLOS Pathogens

Xavier Nassif

Section Editor

PLOS Pathogens

Kasturi Haldar

Editor-in-Chief

PLOS Pathogens

orcid.org/0000-0001-5065-158X

Michael Malim

Editor-in-Chief

PLOS Pathogens

orcid.org/0000-0002-7699-2064

Reviewer's Responses to Questions

**Part I - Summary**

Reviewer #1: In this article, authors perform a systematic deletion of all known pairs of antibacterial T6SS effectors and cognate immunity proteins (E/I) in Pseudomonas aeruginosa. The contribution of individual effectors was analyzed via intraspecies competition between mutants and wild-type cells by imaging mixed bacterial macrocolonies. It was observed that the susceptibility of preys cells varies according to the effector/immunity pair they lack. Also, the initial composition of the community influences the outcome of the competition, with the frequency of contacts between attacker and prey cells at an initial stage being key to the outcome.

All the key concepts touched in the articles are already established in the T6SS field. The differential of the study is that it included all known P. aeruginosa T6SS effector/immunity pairs described to date. The article mainly expands on the concept described by LaCourse et al Nat Microbiol 2018, which described the difference in effectiveness of individual effectors and their synergistic mode of action.

The methodology chosen to analyze the outcome of the competition (imaging macrocolonies) could have been better explored if authors included some sort of quantification (center and edge of colonies, etc). As the work stands, the outcome if mainly visual and does not bring new insights. The study would have more significance if authors used their E/I mutant strains as attackers in competition with relevant species that are common competitiors of Pseudomonas in natural settings.

There are several key points in experimental settings that could be responsible for the outcomes and were not discussed/considered by the authors.

Reviewer #2: In the manuscript, the authors describe a study on the role of the P. aeruginosa type VI secretion system (T6SS) in interbacterial competition, when attackers (T6SS+) and prey cells interact on agar surface. The P. aeruginosa chromosome carries genes encoding for 3 distinct T6SSs and diverse killing effectors. Using a comprehensive collection of mutants with calibrated expression/delivery of toxins specific for each T6SS, the authors utilized advanced microscopy methods to image bacterial survival and growth in colonies containing attacker/prey pairs of wild type and mutant bacteria. They also show that the community of structure was strongly influenced by the type of toxin, their quantity, or combination delivered to the prey. Finally, the authors concluded the study by demonstrating a role for twitching motility, mediated by type 4 pili, for the prey to escape attackers.

This is an interesting paper with a large amount of data for explaining a complex community behavior of P. aeruginosa, addressing the role of multiple toxin secreted by the T6SS and bacterial motility (although curiously, flagellar motility was not considered).

**Part II – Major Issues: Key Experiments Required for Acceptance**

Reviewer #1: 1) Use/develop some quantitative method to quantify and analyze the outcomes of the mixed macrocolonies (center and edge).

2) Complement all E/I mutants with a plasmid containing the immunity protein in order to restore the resistant phenotype.

3) The fact that their prey E/I mutants are in a different background (repressed T6SS) compared to the attacker WT cells (active T6SS) worries me about the outcomes because the expression of all E/I pairs is probably downregulated in all prey cells. Authors should monitor the expression of all E/I pairs in prey cells and/or perform all the competitions in the same active T6SS background. On this topic, the quantification of secreted individual effectors in attacker cells using western blot is also important as authors claim in the text that the difference between retS, rsmA and rsmA/N mutants is due to the amount of effectors injected, which in my opinion seems to be due to the number of firing events.

4) Authors should discuss and consider that the outcome of the killing and their results is due to the capacity of the prey cells of repair the damage caused by individual toxins, which is directly associated with the growth phase of the individual prey cells.

5) The initial cell density (OD600nm) and ratio between attacker and prey (e.g. 1:2) should be clear in all experiments/figures.

6) Fig 6F needs a control to measure T6SS expression and secretion levels and firing frequency in pilA mutant

Reviewer #2: None

**Part III – Minor Issues: Editorial and Data Presentation Modifications**

Reviewer #1: 1) Fig 1B and C: Change the colours and include patterns to facilitate visualization. Fig. 1D and E: correct the name of the mutant strain tssB. Fig 1F: Include scale bar and phase contrast images in another row or merged.

2) S3Fig: the difference in temperatures is not clear. Label the figures to facilitate.

3) Videos: Include scale bars and time stamps. Labels with strain names will make it easier too!

4) Several supplementary figures (e.g. S5, S9, S10, S11, etc) are low resolution or labeled with a different figure number (probably from a previous submission that was not reformatted properly). Please correct those.

5) Names of mutant strains should be properly written with a delta letter from greek alphabet before the gene name. The way it is written is the text is very confusing and informal (e.g. "retS background").

6) Table S1: gene names should be in italic.

7) Fig 3: there is not A and B as described in the text.

8) Fig S7 and S8 seem to be misplaced.

9) Standardize Gac/Rsm or Gac-Rsm

10) Line 423: through instead of trough

Reviewer #2: 1. The construction of plasmid reporter systems used in SFig 1 is confusing, perhaps because it was described inadequately. It took some time for this reviewer to understand (possibly incorrectly) based on the rather brief description in the Materials and Methods that each construct used a similar the PCR amplicon (+30 to -500) relative to the start codon of tssA genes. These were than fused to the promoterless GFPmut3b  gene. Presumably, one set was fused to the coding sequence of GFP, another set created transcriptional fusions. This should be explained clearly, with appropriate detail, without a reader’s need to consult the primer list in the S3Table. The list of primers also lacks a Legend, making the reader guess about a few features (for example, where is the fusion junction for translational fusions and what is the meaning of (restriction sites) in several descriptions.

Moreover, one can quibble about the terminology “transcriptional/translational” fusion, since translation depends on transcription, and these plasmids did not address the extent of translational regulation at all. Yet the authors claim they can differentiate this, as indicated in the description and interpretation of Figure 2 and S3Figures and lines 119-120).

2. The conclusion that there is a “rheostat-fashion” increase in Hcp1 is not correct, the data simply show additive effects of mutations. A rheostat effect requires demonstration of a gradual effect of change with changes in the amounts of the regulatory protein. Moreover, these effects were obtained using regulatory mutations with pleotropic effects, effecting other functions besides T6SS. This limitation should be acknowledged somewhere, either in the Results or Discussion.

3. Whereas it is clear how intermixing and expansion were determined, a few questions about the experimental setup remain. In most cases, why was not there an increase in the number of attacker cells (labeled with GFP) since the prey cells could not kill them? The exception is Figure 2B where the growth of killer cells at the edge is obvious.

4. The growth of prey cells, as shown by the formation of red sectors in the colony images, is surprising, since they should be killed by the attackers. Did the authors determine the ratio of viable cells (attackers and prey) in each colony? What are the surviving prey cells?

5. The figure legend for S5 Figure is insufficient. What are the two lines and three shaded areas? Why this Figure is also labeled Figure 2-Supplement 2?

6. The relative potency of the various toxins was assessed systematically throughout the paper as was the contribution of different mutations, using colony images. When assessing the relative effect of a mutation or potency of a toxin, some form of quantification is required, including appropriate statistical evaluation of the significance. The quantification of intermixing in one experiment is shown in the S5B Fig. Therefore, it should be possible to evaluate and compare the relative Intermixing Indexes. These data should be shown, since it is hard to evaluate statements, such as “barely restricted” (line 183), “ restricted in growth” (line 184) or referring to the “most potent” toxins (lines 187,191 & 192) and “drastically improve” the impact of toxins (line 190).

PLOS authors have the option to publish the peer review history of their article (what does this mean?). If published, this will include your full peer review and any attached files.

Reviewer #1: No

Reviewer #2: No
---

## [Editor Report · Decision Letter 1]

17 May 2023

Dear Prof. Filloux,

We are pleased to inform you that your manuscript 'Effectiveness of Pseudomonas aeruginosa type VI secretion system relies on toxin potency and type IV pili-dependent interaction' has been provisionally accepted for publication in PLOS Pathogens.

Best regards,

Sophie Helaine

Academic Editor

PLOS Pathogens

Xavier Nassif

Section Editor

PLOS Pathogens

Kasturi Haldar

Editor-in-Chief

PLOS Pathogens

orcid.org/0000-0001-5065-158X

Michael Malim

Editor-in-Chief

PLOS Pathogens

orcid.org/0000-0002-7699-2064

---

## [Editor Report · Acceptance letter]

26 May 2023

Dear Prof. Filloux,

We are delighted to inform you that your manuscript, "Effectiveness of *Pseudomonas aeruginosa* type VI secretion system relies on toxin potency and type IV pili-dependent interaction," has been formally accepted for publication in PLOS Pathogens.

Best regards,

Kasturi Haldar

Editor-in-Chief

PLOS Pathogens

orcid.org/0000-0001-5065-158X

Michael Malim

Editor-in-Chief

PLOS Pathogens

orcid.org/0000-0002-7699-2064